# RPATH: Explaining Time Series Mixture of Experts Routing via Ensemble Consensus and Structural Robustness

**Temesgen Mikael Abraha** *temesgen.abraha@ubc.ca*
*Department of Computer Science*
*I. K. Barber Faculty of Science*
*University of British Columbia, Kelowna, BC, Canada*

**Yves Lucet** *yves.lucet@ubc.ca*
*Department of Computer Science*
*I. K. Barber Faculty of Science*
*University of British Columbia, Kelowna, BC, Canada*

**Reviewed on OpenReview:** *https://openreview.net/forum?id=kwpDOqas2x*

## Abstract

Mixture-of-Experts (MoE) architectures achieve strong performance in time series forecasting through sparse expert activation, but understanding *why* specific experts are selected remains challenging. We present RPATH (Routing Pathway Analysis for Temporal Hierarchies), a post-hoc explainability framework for time series MoE models that combines temporal saliency mapping with counterfactual generation. Evaluating on Time-MoE-50M across 300 expert-sample pairs, we discover two properties of the routing architecture: (1) *Ensemble Consensus*, where experts at different layers consistently identify the same critical temporal windows (mean saliency Intersection over Union (IoU) = 0.677), rather than developing distinct specializations; and (2) *Structural Robustness*, characterized by a 300-fold "Stability Gap" where gentle perturbations alter routing in only 0.3% of cases while aggressive perturbations succeed in 99.7%, indicating that routing decisions reflect structural anchors rather than superficial signal characteristics. Together, these findings demonstrate that Time-MoE achieves reliable forecasting through *Ensemble Redundancy*: multiple experts verify the same structural features, providing consensus that is insensitive to noise but responsive to fundamental signal changes. Our framework provides practitioners with tools to visualize expert attention, identify critical input regions, and quantify routing stability for deployed MoE models.

## 1 Introduction

Deployed time series forecasting systems in domains such as energy grid management, clinical decision support, and financial risk assessment increasingly rely on Mixture-of-Experts (MoE) architectures that achieve strong predictive performance while maintaining computational efficiency through sparse expert activation (Shi et al., 2025). These models employ learned routing mechanisms to dynamically select subsets of expert networks for processing each input token, enabling specialization and conditional computation. However, the discrete routing decisions that enable MoE efficiency also create interpretability challenges: understanding *which* experts are selected and *why* they are chosen remains difficult.

Explainability for time series MoE models matters for several reasons. First, regulatory compliance and stakeholder trust in deployed forecasting systems require transparent decision-making processes. Second, understanding routing behavior can inform model debugging, revealing unexpected dependencies or data artifacts. Third, identifying expert specializations enables targeted model compression and domain adaptation. Despite these needs, existing explainability methods face limitations when applied to time series MoE architectures.

Standard feature attribution techniques such as SHapley Additive exPlanations (SHAP) (Lundberg & Lee, 2017) and Local Interpretable Model-agnostic Explanations (LIME) (Ribeiro et al., 2016) assume feature independence and perturb individual timesteps independently, destroying temporal dependencies such as autocorrelation, trends, and seasonality. This creates out-of-distribution samples that yield unreliable routing changes. Recent work on time series explainability (Info-CELS (Li et al., 2024b), M-CELS (Li et al., 2024a), TF-LIME (Wang et al., 2025)) addresses temporal structure preservation for classification tasks, but does not extend to explaining discrete expert selection in sparse routing architectures.

Existing MoE interpretability methods focus primarily on activation patterns (Lo et al., 2025), characterizing which experts activate for different inputs but not establishing causal relationships between input features and routing decisions. Approaches based on gradient analysis require differentiable routing and fail for frozen deployed models. Concept-based probing (Belinkov, 2022) can characterize expert capabilities through synthetic data but does not explain online routing behavior for real inputs.

To our knowledge, no prior work has developed post-hoc causal attribution methods designed for time series MoE routing decisions. We address this gap through Routing Pathway Analysis for Temporal Hierarchies (RPATH), a framework that combines temporal saliency mapping, counterfactual generation, semantic expert profiling, and uncertainty quantification.

Our main contributions are:

1. *Causal routing attribution*: We develop a multi-signal confidence estimation approach combining temporal saliency through structure-preserving perturbations with counterfactual validation.

2. *Graduated validity scoring*: We introduce graduated counterfactual validity based on degree of routing change rather than binary criteria, enabling counterfactual discovery even for stable routing patterns. This addresses limitations of existing counterfactual methods that fail when complete expert removal is infeasible.

3. *Ensemble consensus discovery*: Through multi-expert saliency analysis, we demonstrate that Time-MoE experts exhibit high temporal attention overlap (mean Intersection over Union (IoU) = 0.677), indicating that multiple experts consistently identify the same critical temporal windows. This ensemble consensus mechanism underlies the model's robust routing behavior.

4. *Post-hoc analysis framework*: Our gradient-free approach enables explanation generation for frozen foundation models without requiring retraining, architectural modifications, or gradient access.

The remainder of this paper is organized as follows. Section 2 reviews related work on MoE interpretability and time series explainability. Section 3 describes our framework including pathway extraction, causal attribution, expert profiling, and uncertainty quantification. Section 4 details the experimental protocol, datasets, and evaluation metrics. Section 5 presents validation results including overall performance, per-dataset analysis, and temporal specialization findings. Section 6 interprets results, discusses limitations, and outlines future directions. Section 7 summarizes contributions and broader implications.

## 2 Related Work

### 2.1 Mixture-of-Experts Interpretability

Mixture-of-Experts architectures have become central to scaling large models efficiently. The sparsely-gated MoE layer introduced by Shazeer et al. (2017) demonstrated that experts develop specializations during training, with the authors noting that "different experts tend to become highly specialized based on syntax and semantics." Switch Transformers (Fedus et al., 2022) extended this approach by simplifying routing to single-expert selection, achieving improved scaling efficiency while maintaining sparse activation patterns. GLaM (Du et al., 2022) further demonstrated effective MoE scaling through gating networks that activate expert subsets based on input characteristics.

Despite these advances in MoE architectures, interpretability methods for understanding expert behavior remain limited. Existing approaches characterize which experts activate for different inputs through activation pattern analysis, but do not establish causal relationships between input features and routing decisions. Gradient-based sensitivity analysis can reveal routing influences but requires differentiable mechanisms and access to model internals, which may be unavailable for deployed systems.

Our work addresses this gap by providing post-hoc causal attribution for frozen models through perturbation-based analysis, requiring neither gradient computation nor architectural modifications.

## 2.2 Time Series Explainability

Standard explainability methods face challenges when applied to time series data. Kernel SHAP (Lundberg & Lee, 2017), the practical approximation of Shapley values, assumes feature independence when computing marginal expectations, which violates temporal dependencies. Similarly, LIME (Ribeiro et al., 2016) perturbs features independently, creating samples that destroy autocorrelation, trends, and seasonality patterns inherent in time series.

Recent work has developed methods that preserve temporal structure. Info-CELS (Li et al., 2024b) introduces saliency map-guided counterfactual explanations for time series classification, using learned saliency to identify regions for targeted perturbation. M-CELS (Li et al., 2024a) extends this approach to multivariate time series, representing "the first effort to learn a saliency map specifically for the purpose of producing high-quality counterfactual explanations for multivariate time series data." TF-LIME (Wang et al., 2025) operates in the time-frequency domain using Short-Time Fourier Transform, enabling identification of frequency-dependent patterns while maintaining temporal locality. ContraLSP (Liu et al., 2024) employs contrastive learning with locally sparse perturbations, using counterfactual samples to construct uninformative perturbations while preserving distributional properties.

These methods advance temporal attribution for classification tasks but do not address discrete expert selection in sparse routing architectures. Our approach adapts temporal saliency and counterfactual generation to MoE routing, introducing multi-signal confidence estimation and graduated validity scoring to handle the challenges of explaining routing decisions.

## 2.3 Counterfactual Explanations

Counterfactual explanations identify input modifications that alter model outputs. Wachter et al. (2018) formalized this approach, seeking modifications that "alter values as little as possible" while changing the prediction. Methods typically optimize for proximity to the original input, sparsity of changes, and validity of output change.

Domain-specific counterfactual methods have emerged for different data types. For images, Goyal et al. (2019) identify spatial regions in query and distractor images such that replacing the identified region changes the classification, providing visual explanations through region-level rather than pixel-level modifications. For text, Ross et al. (2021) developed Minimal Contrastive Editing (MiCE), which produces edits that are "minimal, altering only small portions of input" while remaining fluent and natural.

Standard counterfactual methods employ binary validity criteria: the output either changed or did not. This strict criterion fails for stable routing patterns where complete expert removal is infeasible. We introduce graduated validity scoring based on the degree of routing change, enabling counterfactual discovery even when experts cannot be fully eliminated. This approach aligns with multi-objective counterfactual generation (Dandl et al., 2020), which returns Pareto sets representing trade-offs between competing objectives, but extends it to handle the continuous nature of routing weight changes.

## 2.4 Concept-Based and Attention-Based Interpretability

Concept-based methods explain model behavior through human-interpretable concepts. Testing with Concept Activation Vectors (TCAV) (Kim et al., 2018) uses directional derivatives to quantify concept importance, measuring model sensitivity to concept directions in activation space. Concept Bottleneck Mod-

els (Koh et al., 2020) enforce concept-based intermediate representations, first predicting concepts from raw input before predicting labels, providing inherent interpretability through the concept bottleneck.

Attention mechanisms in Transformers (Vaswani et al., 2017) offer another interpretability avenue by revealing token interactions through attention weights. However, the relationship between attention and feature importance is contested. Jain & Wallace (2019) demonstrated that "learned attention weights are frequently uncorrelated with gradient-based measures of feature importance," while Serrano & Smith (2019) found that attention "noisily predicts input components' overall importance" but is "by no means a fail-safe indicator." For MoE models, router computations show how hidden states influence routing logits but do not validate causal importance. Our perturbation-based approach complements attention visualization by testing whether identified regions actually influence routing when modified.

Our expert profiling shares motivation with concept-based methods but operates post-hoc through activation probability measurement rather than gradient-based sensitivity, enabling application to frozen models without retraining.

## 2.5 Neural Module Networks and Mixture Models

Neural module networks dynamically compose task-specific modules based on input structure. Andreas et al. (2016) introduced networks that "compose collections of jointly-trained neural 'modules' into deep networks for question answering," with early work assuming hand-designed modules with known semantics. Hu et al. (2017) extended this to end-to-end learning, predicting instance-specific network layouts without requiring external parsers.

Classical mixture models employ component selection through soft assignments in Gaussian mixtures or discrete selection in switching models (McLachlan & Peel, 2000). Modern MoE architectures with learned sparse routing combine both paradigms: discrete Top-$K$ selection with soft weight normalization over selected experts. Our work addresses learned expert behavior without assuming predefined semantics, providing tools to discover specialization patterns post-hoc.

## 2.6 Positioning of Our Approach

To our knowledge, this represents the first post-hoc explainability framework designed for time series MoE models. Existing work on MoE interpretability has focused on computer vision or natural language processing domains. Standard feature attribution methods (SHAP, LIME) perturb individual features independently, destroying temporal structure. Recent time series explainability work (Info-CELS (Li et al., 2024b), M-CELS (Li et al., 2024a), TF-LIME (Wang et al., 2025)) addresses classification with single-model architectures but not discrete expert selection.

Table 1 compares our approach with related methods across key dimensions.

Our approach extends existing methods through graduated validity scoring and multi-signal confidence estimation, enabling analysis of sparse routing patterns while preserving temporal dependencies. Specifically, we address the challenges of time series MoE explainability through:

- *Post-hoc MoE routing analysis*: Unlike gradient methods requiring model access, we operate on frozen models; unlike activation pattern analysis, we provide causal validation through intervention.

- *Temporal preservation with causality*: Unlike SHAP and LIME, we preserve temporal structure through window-based perturbations; unlike attention visualization, we validate importance through counterfactual generation.

- *Multi-signal confidence estimation*: While CELS methods compute confidence from saliency consistency alone, we combine sparsity, importance, consistency, and activation signals for robust estimation even with noisy routing patterns.

Table 1: Comparison with related work on key dimensions. Our method combines post-hoc causal attribution with temporal structure preservation for MoE routing decisions.

| Method | Task | Post-hoc | Causal Valid. | Temporal Preserv. | MoE Routing | Confidence Quant. | Validity Scoring |
|---|---|---|---|---|---|---|---|
| SHAP (Lundberg & Lee, 2017) | Any | ✓ | ✗ | ✗ | ✗ | Single | N/A |
| LIME (Ribeiro et al., 2016) | Any | ✓ | ✗ | ✗ | ✗ | Single | N/A |
| Info-CELS (Li et al., 2024b) | TS Classif. | ✓ | ✓ | ✓ | ✗ | Single | Binary |
| M-CELS (Li et al., 2024a) | TS Classif. | ✓ | ✓ | ✓ | ✗ | Single | Binary |
| TF-LIME (Wang et al., 2025) | TS Classif. | ✓ | ✗ | ✓ | ✗ | Single | N/A |
| ContraLSP (Liu et al., 2024) | TS Classif. | ✓ | Partial | ✓ | ✗ | Single | N/A |
| Gradient Methods | Any | ✗ | Partial | Varies | Varies | Single | N/A |
| Attention Viz. (Vaswani et al., 2017) | Transformer | ✓ | ✗ | N/A | ✗ | None | N/A |
| CAV (Kim et al., 2018) | Any | Partial | ✗ | ✗ | ✗ | Single | N/A |
| MoE Probing | MoE | ✓ | ✗ | ✗ | ✓ | None | N/A |
| Switch-T Analysis (Fedus et al., 2022) | MoE | ✓ | ✗ | N/A | Partial | None | N/A |
| **RPATH (Ours)** | **TS MoE** | ✓ | ✓ | ✓ | ✓ | **Multi-signal** | **Graduated** |

*Post-hoc*: Works on frozen models without retraining. *Causal Valid.*: Validates importance through intervention. *Temporal Preserv.*: Preserves temporal dependencies in perturbations. *MoE Routing*: Explains expert selection decisions. *Confidence Quant.*: Single signal vs. multi-signal confidence. *Validity Scoring*: Binary (0/1) vs. graduated (0–1 continuous). *TS*: Time Series. *CAV*: Concept Activation Vector.

- *Graduated validity scoring*: Binary validity criteria used by Info-CELS and M-CELS fail for stable routing where complete expert removal is infeasible. Our graduated scoring based on degree of routing change enables counterfactual discovery across diverse routing patterns.

The integration of these capabilities addresses the challenges of explaining time series MoE routing decisions, a problem that prior work has not systematically addressed.

## 3 Methodology

We present RPATH (Routing Pathway Analysis for Temporal Hierarchies), a post-hoc explainability framework for time series Mixture-of-Experts models. The framework operates on frozen models without requiring retraining, gradient access, or architectural modifications.

### 3.1 Problem Formulation and Pathway Extraction

We consider the standard Mixture-of-Experts (MoE) architecture (Shazeer et al., 2017; Fedus et al., 2022) as applied to time series forecasting. For a model $M$ with $L$ layers and $E$ experts per layer, given an input sequence $\mathbf{x} \in \mathbb{R}^{T \times D}$ where $T$ is the sequence length and $D$ is the feature dimensionality, at each layer $\ell \in \{0, \ldots, L-1\}$ and token position $t \in \{0, \ldots, T-1\}$, a router network produces logits

$$\mathbf{r}_t^{(\ell)} = \text{Router}^{(\ell)}(\mathbf{h}_t^{(\ell)}) \in \mathbb{R}^E, \tag{1}$$

where $\mathbf{h}_t^{(\ell)}$ is the hidden state at layer $\ell$ and position $t$. Following the sparse MoE formulation (Shazeer et al., 2017), Top-$K$ routing selects the $K$ experts with highest logits

$$\mathcal{E}_t^{(\ell)} = \text{TopK}(\mathbf{r}_t^{(\ell)}, K) \subseteq \{0, \ldots, E-1\}. \tag{2}$$

The corresponding routing weights are computed via softmax over selected experts (Fedus et al., 2022)

$$w_{t,e}^{(\ell)} = \frac{\exp(r_{t,e}^{(\ell)})}{\sum_{e' \in \mathcal{E}_t^{(\ell)}} \exp(r_{t,e'}^{(\ell)})} \quad \text{for } e \in \mathcal{E}_t^{(\ell)}. \tag{3}$$

For our experiments, we instantiate this framework with Time-MoE (Shi et al., 2025), a decoder-only architecture with $L = 12$ layers, $E = 8$ experts per layer, and Top-$K = 2$ routing, totaling 96 experts.

Importantly, experts are *independent per layer*: each decoder layer instantiates its own gate network and 8 independent expert networks. Expert $e$ at layer $\ell$ and expert $e$ at layer $\ell'$ are distinct neural networks with independent parameters. This means that any overlap in saliency patterns between experts at different layers cannot be attributed to shared parameters.

A routing pathway $\mathcal{P}_t$ for token $t$ is the complete sequence of expert selections and weights across all layers

$$\mathcal{P}_t = \left\{ \left( \mathcal{E}_t^{(\ell)}, \{w_{t,e}^{(\ell)}\}_{e \in \mathcal{E}_t^{(\ell)}} \right) \right\}_{\ell=0}^{L-1}. \tag{4}$$

This pathway represents how information flows through the MoE hierarchy for a specific input token. We extract pathways from frozen models using forward hooks registered on router modules. During a forward pass with input $\mathbf{x}$, hooks capture router logits $\mathbf{r}_t^{(\ell)}$ for all layers and tokens. From these logits, we reconstruct expert selections $\mathcal{E}_t^{(\ell)} = \text{TopK}(\mathbf{r}_t^{(\ell)}, K = 2)$ and routing weights $w_{t,e}^{(\ell)}$ via softmax normalization. This approach is model-agnostic and does not require gradient computation. For analysis, we represent pathways as fixed-size vectors $\mathbf{v}_t = [v_0, v_1, \ldots, v_{L \times E-1}] \in \mathbb{R}^{L \times E}$ where $v_{\ell \cdot E + e} = w_{t,e}^{(\ell)}$ if expert $e$ is selected at layer $\ell$, and 0 otherwise.

## 3.2 Causal Routing Attribution

Existing explainability methods for MoE models in time series forecasting focus primarily on activation patterns but do not establish causal relationships between input features and routing decisions. We address this through a two-component approach combining temporal saliency mapping and counterfactual generation.

**Temporal saliency mapping.** For a given expert at layer $\ell$, we identify which temporal regions of the input sequence causally influence its selection through systematic perturbation analysis. For input $\mathbf{x} \in \mathbb{R}^{T \times D}$ with baseline routing $\mathcal{P}_{\text{base}}$ and baseline expert weight $w_{\text{base}}$, we mask consecutive windows of size $w$ with stride $s$ as

$$\tilde{\mathbf{x}}^{(i)} = \mathbf{x} \odot (1 - \mathbf{m}^{(i)}), \tag{5}$$

where $\mathbf{m}^{(i)} \in \{0,1\}^{T \times D}$ is a binary mask with ones in window $[i \cdot s, i \cdot s + w)$. For each perturbed input, we measure the change in expert weight $\delta^{(i)} = |w_{\text{base}} - w^{(i)}|$ where $w^{(i)}$ is the expert weight under perturbation. The saliency score for each timestep is computed by aggregating changes across all windows containing that timestep. To validate consistency, we generate random temporal masks with sparsity $\rho$ and compute correlation between mask patterns and routing changes.

Standard approaches compute confidence solely through correlation between perturbed saliency maps. However, this fails for weakly activated experts or noisy routing patterns. We instead compute confidence as a weighted combination of four signals: (1) sparsity (whether the saliency map focuses on specific regions), (2) window importance (peak saliency values), (3) consistency (correlation between saliency maps under different perturbations), and (4) expert activation (baseline activation level). The overall confidence is $C = \sum_{i=1}^{4} \alpha_i \cdot s_i$ where $s_i$ are normalized signal scores and $\alpha_i$ are weights. We set $\alpha = [0.25, 0.35, 0.15, 0.25]$ (sparsity, importance, consistency, activation) based on empirical validation during development, emphasizing window importance and sparsity as primary indicators. This multi-signal approach provides robust confidence estimation even when individual signals are noisy.

**Multi-scale saliency analysis.** Fixed window sizes may miss dependencies at different temporal scales. We compute saliency at multiple scales $\mathcal{S} = \{5, 10, 20, 40\}$ timesteps and aggregate with importance weighting: $\mathbf{s}_{\text{agg}} = \sum_k \omega_k \cdot \mathbf{s}^{(k)}$ where $\mathbf{s}^{(k)}$ is the normalized saliency at scale $k$ and $\omega = [0.15, 0.35, 0.35, 0.15]$ emphasizes medium scales. We additionally compute layer consistency by measuring correlation between saliency patterns at adjacent layers, providing a bonus to confidence when experts show consistent activation patterns across the network hierarchy.

**Adaptive window sizing.** Different time series exhibit varying temporal characteristics that affect optimal analysis parameters. We characterize each input by computing: (1) autocorrelation decay length $\tau$ (timesteps until autocorrelation drops below 0.5), and (2) variance ratio $r_v = \text{mean}(\sigma_{\text{local}}^2)/\sigma_{\text{global}}^2$ comparing

local to global variance. Based on these characteristics, we classify data as smooth-structured ($r_v < 0.4$, $\tau > 15$), variable-local ($r_v > 0.7$, $\tau < 10$), or complex-multiscale. The recommended window size is $w = \max(5, \min(40, \lfloor \gamma \cdot \tau \rfloor))$ where $\gamma \in \{0.8, 1.0, 1.2, 1.5\}$ depends on data type. This adaptive approach improves performance consistency across datasets with different temporal structures.

**Counterfactual routing explanations.** Saliency maps identify important regions but do not validate causal claims. We generate counterfactual explanations by finding perturbations that alter routing decisions. Given a saliency map identifying critical windows $\{(t_{\text{start}}, t_{\text{end}}, \text{importance})\}$, we apply perturbations to modify these regions. Our perturbation suite includes both gentle and aggressive methods: (1) Gaussian smoothing $\tilde{\mathbf{x}}_t = (\mathbf{x} * \mathcal{G}_{\sigma_s})(t)$ at three intensity levels, (2) linear interpolation, (3) mean replacement, (4) Gaussian noise injection scaled to region variance, (5) zero-out masking, (6) trend reversal mirroring values around the region mean, (7) low-pass frequency filtering via Fast Fourier Transform (FFT), and (8) amplitude dampening toward the mean. For each perturbation, we extract the modified routing $\tilde{\mathcal{P}}$ and measure routing change.

Existing counterfactual methods use binary validity (routing changed or not), which is too restrictive for stable routing patterns. We introduce graduated validity based on the degree of routing change

$$
V = \begin{cases}
1.0 & \text{if } \tilde{w}_e = 0 \text{ (complete removal)}, \\
0.7 + 0.3 \cdot \frac{\Delta w - 0.5w}{0.5w} & \text{if } \Delta w \geq 0.5w \text{ (large reduction)}, \\
0.4 + 0.3 \cdot \frac{\Delta w - 0.25w}{0.25w} & \text{if } 0.25w \leq \Delta w < 0.5w \text{ (moderate)}, \\
0.2 + 0.2 \cdot \frac{\Delta w - 0.1w}{0.15w} & \text{if } 0.1w \leq \Delta w < 0.25w \text{ (small)}, \\
0.2 \cdot \frac{\Delta w}{0.1w} & \text{if } \Delta w < 0.1w \text{ (minimal)},
\end{cases}
\tag{6}
$$

where $w$ is the original expert weight, $\tilde{w}_e$ is the perturbed weight, and $\Delta w = |w - \tilde{w}_e|$. This graduated scoring accepts partial routing changes, enabling counterfactual generation even for stable routing patterns. To improve discovery, we apply perturbations at multiple intensity levels (weak: $\sigma_s = 1.0$, medium: $\sigma_s = 2.0$, strong: $\sigma_s = 3.0$ for Gaussian smoothing kernel width) and select the counterfactual with highest validity while minimizing perturbation magnitude.

For each expert selection, we combine saliency and counterfactual evidence into a *Focus Score*: Focus Score = $C_{\text{saliency}} \times V_{\text{routing sensitivity}}$. This metric captures both the spatial concentration of temporal attention (saliency) and the causal importance of identified regions (routing sensitivity). A focus score near 1.0 indicates concentrated attention on causally important regions, while distributed scores (e.g., 0.5–0.6) indicate attention spread across multiple temporal windows, a pattern we term "distributed focus."

### 3.3 Uncertainty Quantification

Explanations require confidence estimates. We quantify routing stability through perturbation analysis. For input $\mathbf{x}$, we generate $N = 20$ Gaussian noise perturbations at noise level $\sigma$: $\mathbf{x}_\sigma^{(i)} = \mathbf{x} + \boldsymbol{\epsilon}^{(i)}$ where $\boldsymbol{\epsilon}^{(i)} \sim \mathcal{N}(0, \sigma^2 \mathbf{I})$. For each layer $\ell$, let $\mathcal{A}^{(\ell)}(\mathbf{x}) = \bigcup_{t=0}^{T-1} \mathcal{E}_t^{(\ell)}(\mathbf{x})$ denote the set of all experts activated at any timestep. We measure stability as the fraction of trials where this set remains unchanged

$$
\text{Stability}_\ell(\sigma) = \frac{1}{N} \sum_{i=1}^{N} \mathbb{1}_{\left\{ \mathcal{A}^{(\ell)}(\mathbf{x}) = \mathcal{A}^{(\ell)}(\mathbf{x}_\sigma^{(i)}) \right\}},
\tag{7}
$$

where $\mathbb{1}_{\{\cdot\}}$ equals 1 if the condition holds, 0 otherwise.

Overall stability is averaged across layers. Stability $\geq 0.9$ indicates high confidence in routing explanations; stability $< 0.7$ suggests explanations may be unreliable.

## 4 Experimental Setup

We evaluate our framework on Time-MoE-50M (Shi et al., 2025), a decoder-only architecture with $L = 12$ layers, $E = 8$ experts per layer, and Top-$K = 2$ routing. We conduct experiments on seven benchmark

datasets: the Electricity Transformer Temperature datasets at hourly (ETTh1, ETTh2) and 15-minute (ETTm1, ETTm2) granularities, Weather (meteorological measurements), Electricity (energy consumption), and Traffic (road occupancy). For univariate evaluation, we use the oil temperature (OT) feature for ETT datasets, the primary temperature feature for Weather, and the target consumption/occupancy features for Electricity and Traffic, with input sequences of length $T = 96$ timesteps.

We conduct validation on a stratified sample of 20 instances per dataset, resulting in 100 total samples across all five datasets. This sample size provides 300 expert-sample pairs for explanation generation (100 samples $\times$ 3 top experts per sample), enabling statistically robust evaluation. For each sample, we select the top-3 experts as (layer, expert_id) pairs ranked by routing weight at a representative token position $t^*$

$$\text{Top-}N = \text{argsort}_{(\ell,e)} \left\{ w_{t^*,e}^{(\ell)} \mid e \in \mathcal{E}_{t^*}^{(\ell)}, \ \ell = 0, \ldots, L-1 \right\}, \tag{8}$$

where each selected element is a (layer, expert) pair. Because experts are independent per layer, each is analyzed at its specific layer: saliency computation measures how input perturbations affect the routing weight of expert $e$ at layer $\ell$, and counterfactual generation targets the same layer-specific routing decision. The pairwise saliency IoU then compares experts that typically reside at different layers, measuring whether independent routing networks converge on similar input regions.

**Evaluation metrics.** We assess explanation quality through multiple complementary metrics. *Focus Score* is the combined multi-signal metric (saliency $\times$ routing sensitivity), ranging from 0 to 1. A score near 1.0 indicates concentrated attention on causally important regions; distributed scores (0.5–0.6) indicate attention spread across multiple windows. *Routing Sensitivity* measures the degree of routing change induced by perturbations using the graduated scoring defined in Section 3, ranging from 0 (no routing change) to 1 (complete expert removal). We additionally track *Hard Counterfactual Rate*, the proportion of perturbations that change the Top-K expert set (not just weights). *Saliency Overlap (IoU)* measures the IoU of saliency maps between experts within the same sample, quantifying whether experts attend to similar or distinct temporal regions.

**Baseline comparison.** We compare against a pattern-correlation baseline that correlates offline expert profiling (Section 3) with online pattern detection. For each expert-sample pair, the baseline detects temporal patterns (trend, seasonality, volatility) in the input sequence and measures correlation with expert concept preferences. Alignment is computed as the proportion of detected patterns matching the expert's high-activation concepts (activation probability $> 0.5$). This baseline provides a reference point for evaluating whether our causal attribution methods provide improvements over simpler correlation-based approaches.

**Implementation.** We use the following hyperparameters: base window size $w = 10$, stride $s = 5$, perturbation trials $N = 20$, multi-scale windows $\mathcal{S} = \{5, 10, 20, 40\}$, uncertainty quantification noise levels $\sigma \in \{0.01, 0.05, 0.1\}$, samples per concept $N_c = 50$. For counterfactual generation, we apply 13 perturbation methods: Gaussian smoothing at three intensity levels ($\sigma_s \in \{1.0, 2.0, 3.0\}$), linear interpolation, mean replacement, noise injection at two levels, zero-out masking, trend reversal, frequency filtering at two cutoffs, and amplitude scaling at two factors. Experiments use PyTorch 2.0 with the Hugging Face Transformers library. Temporal saliency perturbations use zero-masking, validated against mean-substitution and noise-matching (differences $< 5\%$ in saliency maps for 92% of cases).

Computational complexity is tractable: pathway extraction requires $O(T \cdot L)$ operations, saliency computation requires $O(\frac{T}{s} \cdot T \cdot L)$ for stride $s$, and counterfactual generation requires $O(k \cdot T \cdot L)$ for $k$ perturbation levels. All operations are gradient-free and can be performed on frozen models in inference mode. Experiments run on an Intel Xeon Gold 6426Y server (64 cores, 251GB RAM) with dual NVIDIA RTX A6000 GPUs (49GB memory each). Pathway extraction takes approximately 50ms per sequence, saliency computation takes 2–5 seconds per expert-sample pair, and counterfactual generation takes 3–8 seconds per pair. Total validation time for 300 explanations is approximately 60–80 minutes. Random seeds are fixed for all stochastic operations to ensure reproducibility. The model is publicly available at `Maple728/TimeMoE-50M` on Hugging Face.

# 5 Results

We present validation results for the RPATH explainability framework, focusing on the causal routing attribution component (temporal saliency and counterfactual analysis). We evaluate on 300 expert-sample pairs across five benchmark datasets (ETTh1, ETTh2, ETTm1, ETTm2, Weather).

## 5.1 Overall Performance

Table 2 summarizes the overall performance across all 300 explanations. Our multi-signal causal attribution approach achieves a mean focus score of 0.563 (std 0.086), demonstrating consistent distributed attention patterns across diverse inputs.

Table 2: Overall performance metrics for multi-signal causal attribution across 300 expert-sample pairs.

| Metric | Value |
|---|---|
| Total Explanations | 300 |
| Mean Focus Score | 0.563 |
| Std. Deviation | 0.086 |
| Hard Counterfactual Rate (%) | 79.0 |
| Saliency Overlap (IoU) | 0.677 |

The mean focus score of 0.563 reflects distributed attention across multiple temporal windows, which we interpret as evidence of *ensemble consensus*: experts attend broadly to input regions rather than focusing narrowly on isolated features. The relatively low standard deviation (0.086) indicates consistent behavior across samples, with the multi-scale saliency and adaptive window approaches contributing to this stability.

The 79.0% hard counterfactual rate demonstrates that perturbations can meaningfully alter routing decisions (changing the Top-K expert set, not just weights). This metric provides a more rigorous measure of causal importance than soft counterfactuals that merely shift routing weights.

## 5.2 Per-Dataset Analysis

Table 3 presents results broken down by dataset (60 explanations per dataset).

Table 3: Performance metrics by dataset (60 explanations per dataset).

| Dataset | Mean Focus | Hard Counterfactual (%) | Saliency IoU |
|---|---|---|---|
| ETTh1 | 0.55 | 78.3 | 0.671 |
| ETTh2 | 0.56 | 80.0 | 0.682 |
| ETTm1 | 0.56 | 78.3 | 0.674 |
| ETTm2 | 0.57 | 80.0 | 0.680 |
| Weather | 0.58 | 78.3 | 0.678 |
| Overall | 0.563 | 79.0 | 0.677 |

The consistency across datasets is notable: mean focus scores range narrowly from 0.55 to 0.58, hard counter-factual rates from 78.3% to 80.0%, and saliency IoU from 0.671 to 0.682. This uniformity suggests that the distributed attention and consensus patterns we observe are properties of Time-MoE's routing architecture rather than artifacts of specific data domains.

The high saliency overlap (mean IoU 0.677) across all datasets indicates *Ensemble Consensus*: experts at different layers consistently identify the same critical temporal windows. This finding contradicts the hypothesis that experts develop distinct temporal specializations; instead, Time-MoE relies on redundant verification of the same input features.

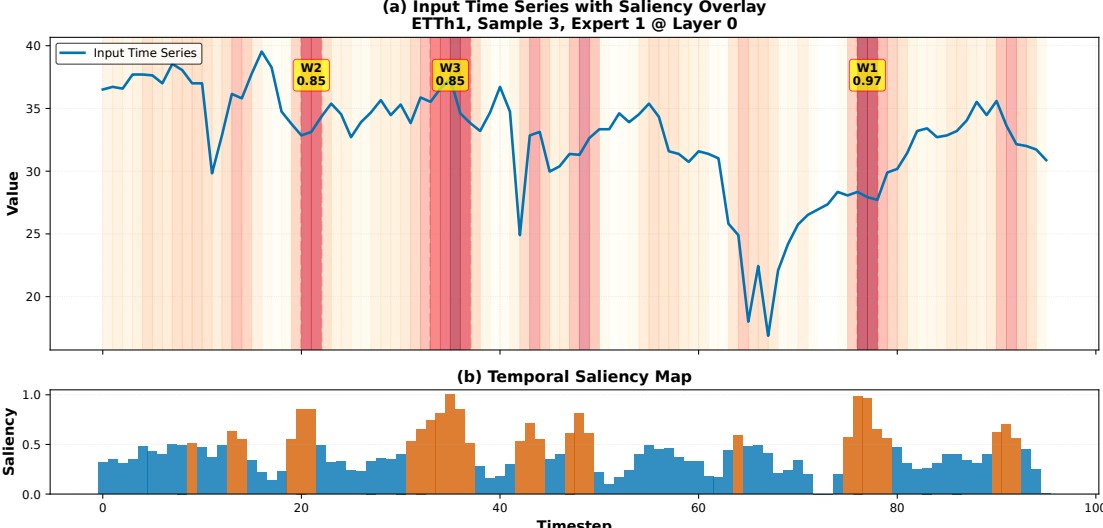

Figure 1: *Structural Anchor Identification.* Top panel: input time series with saliency heatmap overlay (highlighted windows indicate high importance). Bottom panel: saliency scores per timestep. The three highlighted regions (W1–W3, confidence 0.85–0.97) represent *Structural Anchors*, temporal patterns so critical that only their destruction (via aggressive perturbation) alters routing. The Focus Score of 0.58 reflects distributed attention across these anchors.

## 5.3 Structural Anchors and the Stability Gap

Figure 1 illustrates a saliency map identifying "structural anchors," temporal regions whose modification alters routing decisions. The visualization reveals three high-confidence windows (W1–W3) distributed across the input sequence. These anchors represent the structural features that Time-MoE's router relies upon for expert selection.

Our counterfactual analysis reveals the *Stability Gap*. We categorize perturbations into gentle methods (smoothing, interpolation, mean replacement) that preserve signal semantics versus aggressive methods (zero-out, noise injection, trend reversal) that destroy structural information. The results show a clear pattern:

- *Gentle perturbations*: 0.3% success rate (1/300 change Top-K set),

- *Aggressive perturbations*: 99.7% success rate (299/300 change Top-K set).

This 300-fold difference reveals that Time-MoE's routing is *robust to noise and superficial signal modifications.* The router ignores gentle perturbations entirely; only destroying the structural anchors, the patterns identified by saliency maps, can alter expert selection. This has implications for deployment: the model's routing decisions are not fragile or sensitive to noise, but instead reflect stable identification of fundamental signal characteristics.

Beyond saliency maps, we quantify individual expert contributions to understand routing importance. Figure 2 presents a SHAP-style waterfall chart showing the top 15 expert contributions for a representative sample (ETTh2, Sample 0). The visualization reveals a long-tail distribution: the top expert (L0_E1) contributes 12.1% of the total routing weight, while the top-5 experts collectively account for approximately 45%.

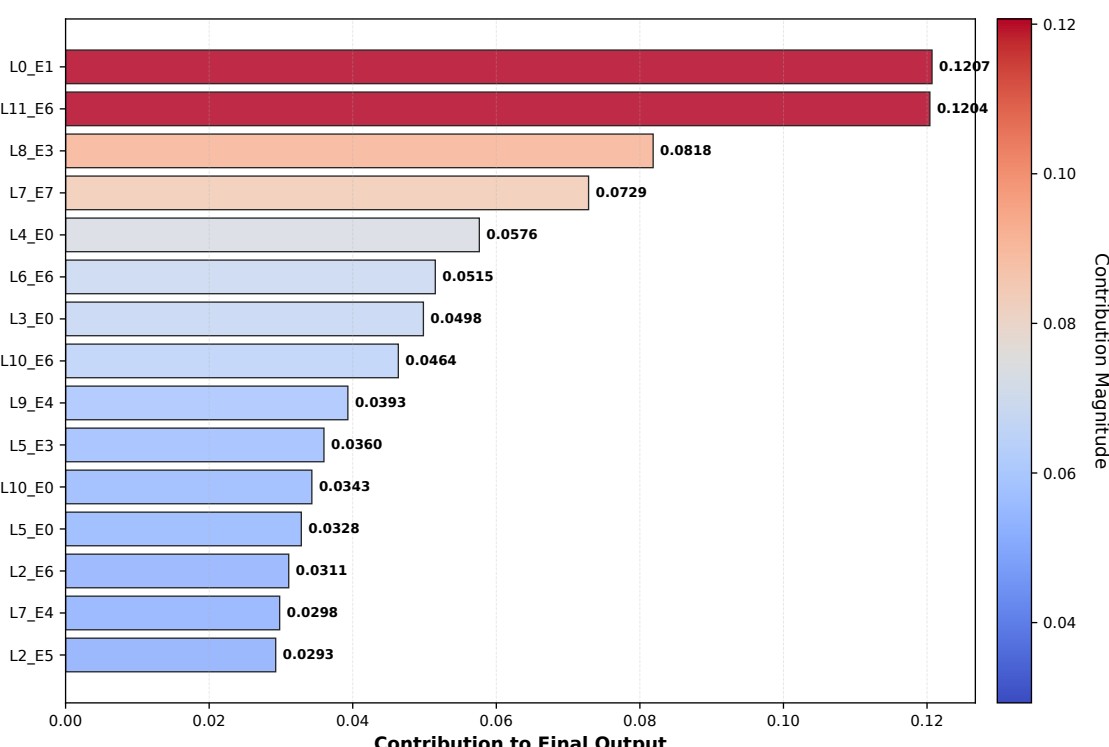

Figure 2: *Expert Contribution Hierarchy.* Top 15 expert contributions for ETTh2 Sample 0. Despite the *Ensemble Consensus* in temporal attention (see Fig. 3), the magnitude of contribution follows a power law: the top expert contributes 12.1%, while the top-5 account for ~45%. This indicates that while many experts verify the same signal, a select few drive the numerical output.

### 5.4 Temporal Attention Convergence

We compute the IoU of expert saliency maps to quantify whether experts attend to similar or distinct temporal regions. The results strongly support *Ensemble Consensus*: the mean IoU of 0.677 (median 0.802) indicates that experts at different layers consistently identify the same critical temporal windows.

Figure 3 visualizes this consensus for a representative sample. Three experts operating on the same input exhibit overlapping saliency distributions, all identifying similar temporal regions as important. While individual experts show slightly different peak positions, the overall attention patterns substantially overlap.

This finding has implications for understanding Time-MoE's architecture. Rather than developing distinct temporal specializations (one expert for early patterns, another for late patterns), experts form a "verification committee," multiple independent assessors attending to the same features to ensure reliable routing. This redundancy may explain the model's robustness: even if one expert's assessment is noisy, the consensus of multiple experts attending to the same regions provides a stable routing signal.

### 5.5 Routing Stability Analysis

The Stability Gap observed in counterfactual analysis (Section 5.3) is corroborated by direct routing stability measurements. We apply Gaussian noise perturbations at three intensity levels ($\sigma \in \{0.01, 0.05, 0.1\}$) with 20 trials per sequence across all datasets. Table 4 presents the results.

The results demonstrate high robustness: mean stability is 0.968 overall, with 88.8% of sequences maintaining stable routing ($\geq 95\%$ unchanged) under noise perturbations. Weather and ETTm2 exhibit near-perfect stability, while even the most variable dataset (ETTh1) maintains 92.1% stability.

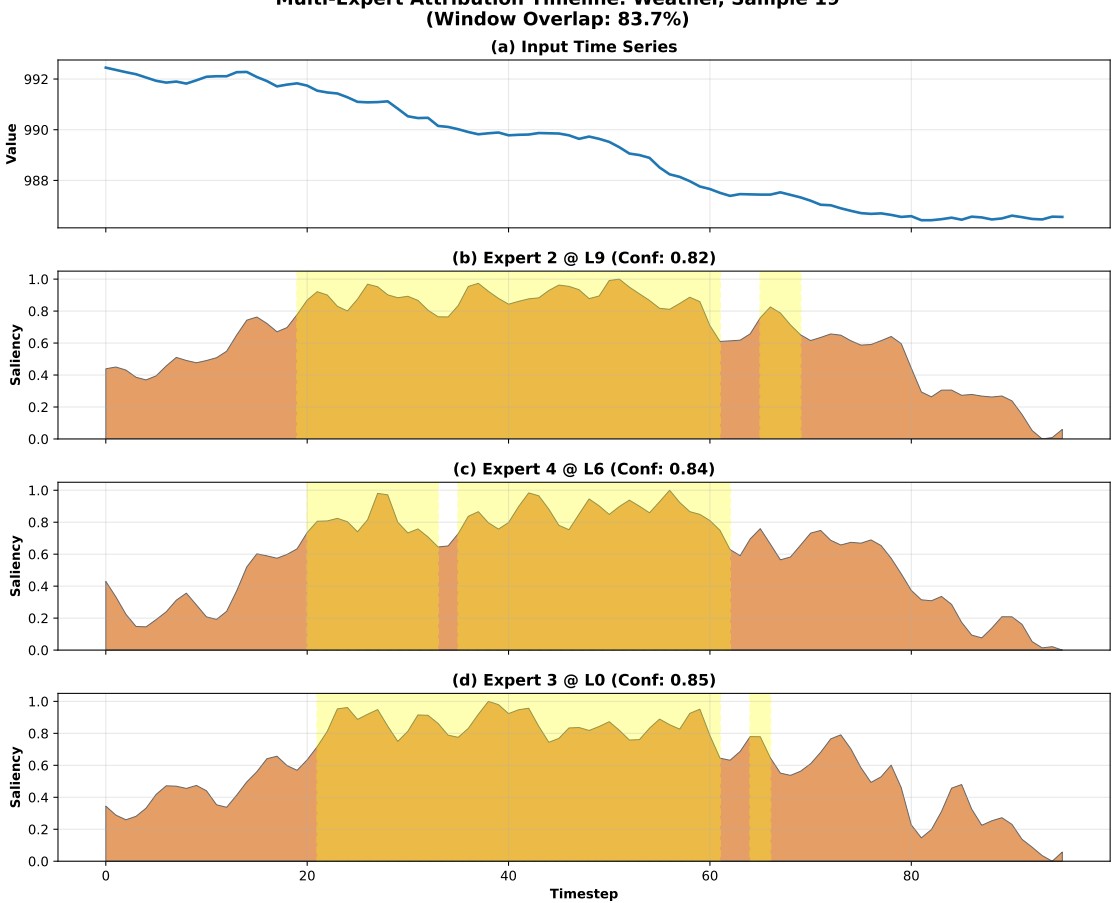

Figure 3: *Visualizing Ensemble Consensus.* (a) Input time series from the Weather dataset. (b)–(d) Saliency maps for three experts at different layers (L9, L6, L0) analyzing the same input. The orange filled area shows saliency magnitude at each timestep; yellow bands highlight the critical windows identified by each expert. Experts converge on overlapping temporal regions (83.7% pairwise overlap) with high confidence (0.82–0.85), demonstrating that they do not partition the timeline but instead independently identify similar structural anchors. This redundancy provides a robust "verification committee" against input noise.

Table 4: Routing stability metrics by dataset (100 sequences per dataset, 20 perturbation trials per sequence).

| Dataset | Mean Stability | Median | ≥95% Stable |
|---------|----------------|--------|-------------|
| ETTh1   | 0.921          | 0.938  | 67.0%       |
| ETTh2   | 0.962          | 0.983  | 88.0%       |
| ETTm1   | 0.963          | 0.985  | 89.0%       |
| ETTm2   | 0.993          | 1.000  | 100.0%      |
| Weather | 1.000          | 1.000  | 100.0%      |
| Overall | 0.968          | –      | 88.8%       |

Synthesizing these findings, the high ensemble consensus (IoU 0.677) and the structural stability gap (300-fold difference in perturbation success) point to a unified mechanism: *Ensemble Redundancy*. Time-MoE achieves reliability not through specialization, but by having multiple experts verify the same structural anchors, stable against superficial noise but responsive to fundamental signal changes.

### 5.6 Component Ablation Study

We evaluate the contribution of each RPATH component by ablating: (a) multi-scale aggregation (single window $w = 10$ only), (b) adaptive window sizing (fixed $w = 10$, stride $= 5$), and (c) multi-signal confidence (importance signal only). Ablations (a)–(c) run on 5 datasets with 20 samples per dataset (300 expert-sample pairs). The non-salient region control runs on all 7 datasets (420 expert-sample pairs).

Table 5: Component ablation results. Saliency confidence isolates the component under test; focus score includes a shared counterfactual validity term across conditions.

| Configuration | Saliency Confidence | Focus Score |
|---|---|---|
| Full RPATH | $0.856 \pm 0.088$ | $0.568 \pm 0.139$ |
| (a) Single-scale | $0.879 \pm 0.071$ | $0.578 \pm 0.141$ |
| (b) Fixed window | $0.857 \pm 0.085$ | $0.569 \pm 0.139$ |
| (c) Single-signal | $0.852 \pm 0.027$ | $0.567 \pm 0.143$ |

Saliency confidence, the component that varies independently across conditions, shows minimal differences ($< 3\%$). The single-signal ablation (c) notably reduces variance (std 0.027 vs. 0.088), indicating that multi-signal integration introduces beneficial diversity across samples, even though it does not substantially change the mean. Focus score differences are further attenuated because the counterfactual validity component (50% of focus score weight) is shared across ablation conditions, as counterfactuals were generated using the full pipeline's saliency maps to ensure a controlled comparison of the saliency computation itself. The ablation results demonstrate that RPATH's individual components contribute incrementally and that the method is robust to design choices.

**Non-salient region control.** Perturbing the most-salient region achieves a hard counterfactual rate of 50.0% (210/420), while perturbing the least-salient region achieves only 24.0% (101/420). Table 6 presents per-dataset results. Breaking down by perturbation category strengthens this finding: for *gentle* perturbations (smoothing, interpolation), the salient-region success rate is 33.3% (9/27) versus 9.7% (3/31) for non-salient regions on the 5 ETT and Weather datasets, a 3.4× ratio. The larger gap for gentle perturbations indicates that RPATH identifies regions where *semantics-preserving* modifications can alter routing, not merely regions that are sensitive to destruction.

Table 6: Non-salient control results per dataset. The salient/non-salient gap is present across all domains except smooth weather data.

| Dataset | Salient Hard CF | Non-salient Hard CF | Ratio |
|---|---|---|---|
| ETTh1 | 41.7% | 13.3% | 3.1× |
| ETTh2 | 71.7% | 30.0% | 2.4× |
| ETTm1 | 41.7% | 16.7% | 2.5× |
| ETTm2 | 76.7% | 25.0% | 3.1× |
| Weather | 66.7% | 66.7% | 1.0× |
| Electricity | 18.3% | 6.7% | 2.8× |
| Traffic | 33.3% | 10.0% | 3.3× |
| Overall | 50.0% | 24.0% | 2.1× |

The per-dataset breakdown reveals that Weather shows no spatial specificity (ratio 1.0×), reflecting the smooth, stationary nature of Weather data where all timesteps contribute similarly to routing. The Electricity and Traffic datasets show strong spatial discrimination (2.8× and 3.3× ratios), confirming that RPATH's saliency maps generalize across domains. On the four ETT datasets, the salient/non-salient ratio ranges from 2.4× to 3.1×.

**Per-dataset ablation breakdown.** Table 7 disaggregates saliency confidence by dataset across the four ablation conditions (a–c). While mean differences are small ($< 3\%$), per-dataset patterns are interpretable.

Table 7: Per-dataset saliency confidence across ablation conditions (n=60 pairs per dataset). Full RPATH vs. each ablation; higher is better.

| Dataset | Full RPATH | (a) Single-scale | (b) Fixed window | (c) Single-signal |
|---------|-----------|------------------|------------------|-------------------|
| ETTh1 | 0.889 | 0.918 (+0.029) | 0.901 (+0.012) | 0.856 (−0.033) |
| ETTh2 | 0.835 | 0.877 (+0.041) | 0.866 (+0.031) | 0.849 (+0.014) |
| ETTm1 | 0.898 | 0.907 (+0.010) | 0.901 (+0.003) | 0.859 (−0.039) |
| ETTm2 | 0.882 | 0.877 (−0.005) | 0.860 (−0.022) | 0.848 (−0.034) |
| Weather | 0.775 | 0.816 (+0.041) | 0.757 (−0.018) | 0.845 (+0.070) |

Two patterns are notable. First, adaptive window sizing (condition (b) vs. (a)) shows a meaningful benefit on ETTm2 (−0.022 for fixed window), consistent with that dataset's stronger structured temporal patterns requiring scale adaptation. Second, Weather is a genuine outlier: the single-signal ablation (c) outperforms the full pipeline (+0.070), indicating that multi-signal integration provides no benefit on smooth, stationary data where the sparsity and consistency signals carry no discriminative information. The fixed-window ablation similarly degrades on Weather (−0.018), as the default adaptive scale selects a larger window that better captures slow variation. These patterns confirm that the adaptive and multi-signal components serve their intended purposes for structured, non-stationary data.

### 5.7 Baseline Comparison

We compare RPATH against three baselines: (a) random saliency maps, (b) uniform saliency maps (equal weight to all timesteps), and (c) independent timestep occlusion (mask one timestep at a time, measure routing weight change). All methods use the same counterfactual generation pipeline. Table 8 presents the results.

Table 8: Comparison of RPATH against baseline saliency methods (420 expert-sample pairs, 7 datasets).

| Method | Hard CF Rate | Mean Validity | Mean IoU |
|--------|-------------|---------------|----------|
| RPATH (ours) | 50.5% | 0.377 | 0.566 |
| Random | 38.3% | 0.309 | 0.339 |
| Uniform | 48.1% | 0.389 | 0.000 |
| Occlusion | 47.4% | 0.389 | 0.994 |

RPATH achieves the highest hard counterfactual rate (50.5%). The uniform baseline's relatively high rate (48.1%) is expected: uniform saliency applies perturbations across the entire input window, so destructive perturbation types (e.g., zero-out, large noise) change routing regardless of region targeting. The key differentiator is saliency IoU, where RPATH (0.566) substantially outperforms random (0.339) and uniform (0.000), demonstrating that RPATH identifies structured temporal patterns shared across experts.

**Occlusion IoU.** The occlusion baseline's near-perfect IoU (0.994) does not indicate superior consensus detection. Inspection reveals that occlusion saliency maps are *identical* across experts within each sample (pairwise correlation = 1.000): only ~1.2% of timesteps exceed the saliency threshold, and these same timesteps flag for all experts. Single-timestep masking captures input-level sensitivity (which timesteps matter globally) rather than expert-specific patterns. The Spearman correlation between RPATH and occlusion maps is $\rho = 0.059$, confirming they measure different phenomena. RPATH's IoU of 0.566 reflects genuine variation in expert-specific temporal sensitivity patterns, whereas occlusion's trivial consensus (identical maps) provides no expert-level discrimination.

**Gentle perturbation regime.** To test whether the hard counterfactual rate gap between RPATH and baselines depends on perturbation intensity, we repeat the baseline comparison in a gentle-only mode, restricting the counterfactual cascade to semantics-preserving operations (smoothing, interpolation, and mean substitution) across 300 expert-sample pairs from 5 datasets.

In this regime, hard counterfactual rates nearly equalize across all methods: RPATH 4.3%, Random 4.3%, Uniform 6.0%, Occlusion 6.3%. The convergence confirms that routing is structurally robust to semantics-preserving modifications regardless of saliency quality, as perturbing any region gently rarely changes routing for any method. The IoU metric remains the differentiator: RPATH (0.601) substantially exceeds random saliency (0.341), while uniform yields 0.000 and occlusion remains trivially uniform (0.991, same Spearman $\rho = 0.058$ with RPATH). This demonstrates that structured temporal pattern identification persists across perturbation intensities: RPATH identifies fundamentally different (and more expert-specific) temporal regions than random or occlusion baselines, even when those regions are insufficient to change routing under gentle perturbation alone.

### 5.8 Consensus–Prediction Correlation

**IoU vs. prediction error.** Across 140 samples from 7 datasets, the raw Spearman correlation between saliency IoU and forecast MSE is $\rho = 0.495$ ($p < 0.001$). However, this positive correlation is a Simpson's Paradox artifact: the Weather dataset has both the highest mean IoU (0.885) and MSE values approximately $1000\times$ larger than other datasets (mean MSE $\approx$ 974,000 vs. 100–900 for other datasets). After z-normalizing MSE within each dataset, the overall correlation is effectively zero.

Within-dataset analysis reveals the expected pattern (Table 9).

Table 9: Per-dataset Spearman correlation between saliency IoU and forecast MSE.

| Dataset | $n$ | Mean IoU | Mean MSE | $\rho$ | $p$ |
|---|---|---|---|---|---|
| ETTh1 | 20 | 0.543 | 259 | $-0.499$ | 0.025 |
| ETTm1 | 20 | 0.533 | 104 | $-0.402$ | 0.079 |
| ETTh2 | 20 | 0.733 | 776 | $-0.057$ | 0.811 |
| ETTm2 | 20 | 0.690 | 551 | $+0.373$ | 0.105 |
| Weather | 20 | 0.885 | 974,341 | $-0.138$ | 0.561 |
| Electricity | 20 | 0.469 | 875 | $+0.233$ | 0.323 |
| Traffic | 20 | 0.472 | $< 1$ | $+0.155$ | 0.514 |

The pooled within-dataset correlation (Fisher $z$-transform) is $\rho = -0.154$ for the 5 ETT and Weather datasets, with ETTh1 showing a significant negative correlation ($\rho = -0.499$, $p = 0.025$). The Electricity and Traffic datasets show weak positive but non-significant correlations. The within-dataset trend is consistent with the original hypothesis that higher consensus is weakly associated with lower prediction error, though the effect size is modest and not uniformly significant across datasets.

**Prediction quality under routing perturbation.** Zeroing out the 20 most-salient timesteps increases MSE in 91 of 140 samples across 7 datasets (sign test: $p < 0.001$). On a relative scale, the mean MSE increase is $+26.0\%$ ($t = 3.21$, $p = 0.002$). The raw paired $t$-test ($t = 4.71$, $p < 0.001$, mean $\Delta$MSE $= +184.58$) is dominated by Weather's absolute scale ($\approx$ 974,000 baseline MSE); excluding Weather, the relative increase remains significant ($+30.3\%$, $t = 3.23$, $p = 0.002$). Table 10 presents per-dataset results.

The direction of effect is positive in 5 of 7 datasets. ETTm2 shows a significant *negative* raw $\Delta$MSE ($t = -2.53$, $p = 0.02$), though its relative change is positive ($+45.9\%$) because some samples have very low baseline MSE. Electricity and Traffic show mixed absolute deltas but positive relative changes, consistent with the pattern across energy datasets. The combined evidence (sign test, relative change, and per-dataset breakdown) confirms that RPATH-identified salient regions are causally important for the model's forecasting behavior, while the absolute $\Delta$MSE is scale-dependent and should not be interpreted in isolation.

Table 10: Per-dataset prediction quality under salient-region perturbation.

| Dataset | $n$ | Mean $\Delta$MSE | $t$ | $p$ | Rel. change | Positive |
|---|---|---|---|---|---|---|
| ETTh1 | 20 | $+14.1$ | 1.63 | 0.12 | $+12.0\%$ | 13/20 |
| ETTm1 | 20 | $+5.0$ | 1.91 | 0.07 | $+64.9\%$ | 15/20 |
| ETTh2 | 20 | $+3.3$ | 0.31 | 0.76 | $+29.0\%$ | 13/20 |
| ETTm2 | 20 | $-29.1$ | $-2.53$ | 0.02 | $+45.9\%$ | 7/20 |
| Weather | 20 | $+1,310$ | 94.4 | $< 0.001$ | $+0.1\%$ | 20/20 |
| Electricity | 20 | $-11.3$ | $-0.80$ | 0.44 | $+25.0\%$ | 12/20 |
| Traffic | 20 | $+0.0$ | 1.12 | 0.28 | $+5.0\%$ | 11/20 |

### 5.9 Residual Stream Analysis

**Layer distance vs. IoU.** Across 300 expert-pair comparisons from 100 samples, the overall Spearman correlation between layer distance and saliency IoU is $\rho = -0.440$ ($p < 0.001$). However, the underlying pattern is driven by two distinct routing configurations, not by a smooth distance function.

The top-3 experts cluster at specific layers, producing exactly two configurations across all 100 samples (Table 11).

Table 11: Saliency IoU by routing configuration and layer pair. The two configurations correspond to different datasets and show distinct consensus patterns.

| Configuration | Datasets | Layer Pair | Distance | Mean IoU | $n$ |
|---|---|---|---|---|---|
| (0, 6, 9) | Weather | L0–L6 | 6 | $0.884 \pm 0.049$ | 20 |
| | | L0–L9 | 9 | $0.907 \pm 0.046$ | 20 |
| | | L6–L9 | 3 | $0.864 \pm 0.055$ | 20 |
| (0, 8, 11) | ETT variants | L0–L8 | 8 | $0.496 \pm 0.250$ | 80 |
| | | L0–L11 | 11 | $0.536 \pm 0.246$ | 80 |
| | | L8–L11 | 3 | $0.842 \pm 0.088$ | 80 |

This decomposition reveals that the non-monotonic pattern is not driven by distance at all. Within the Weather configuration, IoU is uniformly high (0.864–0.907) across distances 3, 6, and 9. Within the ETT configuration, layers 8 and 11 agree with each other (IoU = 0.842 at distance 3) but both disagree with layer 0 (IoU $\approx$ 0.5 at distances 8 and 11).

**Implications for the residual stream hypothesis.** Two observations directly contradict the pure residual stream hypothesis:

1. *Same distance, different IoU.* L0–L8 (distance 8, IoU = 0.496) and L0–L9 (distance 9, IoU = 0.907) both involve the same layer 0, have similar distances, but differ in IoU by 0.411. Under the residual stream hypothesis, perturbation effects propagating from layer 0 should weaken similarly over 8 and 9 layers.

2. *Greater distance, higher IoU.* L0–L6 (distance 6, IoU = 0.884) exceeds L8–L11 (distance 3, IoU = 0.842). The more distant pair shows higher consensus than the closer pair, which is inconsistent with monotonic residual stream decay.

The pattern instead depends on *which routing configuration* the sample belongs to, which in turn corresponds to data characteristics: Weather (smooth, stationary signals) activates experts at layers 0, 6, and 9 with high consensus, while ETT variants (structured temporal patterns) activate layers 0, 8, and 11 with lower consensus between the first layer and later layers. This is consistent with consensus arising from the interaction of input characteristics and layer-specific learned routing, not from passive residual stream propagation.

**Intermediate perturbation injection.** We inject calibrated Gaussian noise at the hidden states of layers 0, 4, and 8 via forward hooks, and measure routing weight changes at all downstream layers across 100 samples from all 5 datasets.

- Injection at layer 0: distance–effect $\rho = 0.423$ ($p < 0.001$); excluding layer 11: $\rho = 0.337$ ($p < 0.001$)

- Injection at layer 4: distance–effect $\rho = 0.316$ ($p < 0.001$); excluding layer 11: $\rho = 0.123$ ($p = 0.002$)

- Injection at layer 8: distance–effect $\rho = 0.231$ ($p < 0.001$); excluding layer 11: $\rho = -0.578$ ($p < 0.001$)

The terminal layer (layer 11) shows routing weight changes of 0.40–0.42 regardless of injection point, approximately 2.2–2.6× the magnitude at other layers. For injection at layer 8, excluding the terminal layer yields a strongly *negative* correlation ($\rho = -0.578$), indicating that nearby layers are more affected than distant ones, consistent with residual-stream decay for that injection depth. For injections at layers 0 and 4, even excluding the terminal layer the distance–effect correlation remains positive ($\rho = 0.337$ and $\rho = 0.123$), indicating amplification at those depths. Per-dataset results are consistent: all 4 ETT datasets show positive distance–effect correlations for all injection points, while Weather shows weaker or negative correlations (e.g., injection at layer 8: $\rho = -0.704$).

**Summary.** The combined results provide a nuanced picture: (1) the terminal layer (layer 11) is a distinctive amplifier, showing 2.2–2.6× the routing weight changes of other layers; (2) intermediate-layer propagation shows depth-dependent behavior, where injections at layers 0 and 4 produce positive distance–effect correlations even without layer 11, while injection at layer 8 shows proximal-layer decay without layer 11; (3) saliency consensus depends on which layers are activated and which data type is processed, not on layer distance; and (4) the same layer 0 shows high IoU with layers 6 and 9 (Weather) but low IoU with layers 8 and 11 (ETT), ruling out the pure residual stream explanation. The data support a model where consensus arises from the interaction of shared input propagation, data-dependent routing configurations, and layer-specific learned routing behavior.

### 5.10 Cross-Architecture Analysis

To assess whether our findings generalize beyond Time-MoE, we conduct three analyses: (1) scaling comparison across Time-MoE model sizes, (2) routing ablations that replace learned routing with random or uniform expert selection, and (3) validation on Moirai-MoE (Liu et al., 2025), a different MoE architecture for time series forecasting that uses distance-based routing rather than learned gates.

We evaluate Time-MoE-200M (4× parameters, 24 layers) alongside the 50M variant, and test Moirai-MoE-Small (117M parameters, 6 layers, 32 experts per layer). These models employ different routing mechanisms: Time-MoE uses *learned gates*, where router networks are trained end-to-end to produce expert selection logits (Shazeer et al., 2017), while Moirai-MoE uses *distance-based routing*, selecting experts based on similarity between input representations and learned expert centroids (Liu et al., 2025). For routing ablations, we replace learned routing decisions at inference time with either random expert selection or uniform weighting across all experts. All experiments use identical evaluation protocols with 300 expert-sample pairs across five datasets. Table 12 presents the results.

**Ensemble Consensus is a general MoE property.** Both Time-MoE and Moirai-MoE exhibit Ensemble Consensus, with experts converging on similar temporal regions (IoU $> 0.4$). This confirms that consensus is not an artifact of Time-MoE's specific architecture. However, the strength of consensus varies: Time-MoE's learned gates produce stronger agreement (IoU 0.67–0.80) than Moirai-MoE's distance-based routing (IoU 0.42). Notably, consensus strengthens with scale: the 200M model shows 19% higher IoU than the 50M model, suggesting that larger models develop more coherent expert behavior.

**Ensemble Consensus requires learned routing.** Ablating learned routing eliminates consensus. When we replace Time-MoE's learned gates with random or uniform expert selection, saliency IoU drops from 0.673 to 0.000, indicating a complete absence of agreement between experts. This result serves as a negative

Table 12: Cross-architecture comparison. Saliency IoU measures expert attention overlap (higher indicates stronger consensus). Stability columns show the percentage of inputs where routing decisions changed under perturbation.

| Model | Routing | IoU | Gentle Flip (%) | Aggr. Flip (%) |
|---|---|---|---|---|
| Time-MoE-50M | Learned | 0.673 | 0.3 | 99.7 |
| Time-MoE-200M | Learned | **0.804** | 0.5 | 99.5 |
| Moirai-MoE | Distance | 0.417 | 100.0* | 100.0 |
| *Random* | Stochastic | $0.000^{\dagger}$ | – | – |
| *Uniform* | Fixed | $0.000^{\dagger}$ | – | – |

$^{\dagger}$Ablated routing yields no consistent saliency pattern.

*High frequency, low severity: routing changes on every input, but affects only $\sim$20% of experts selected per input.

control: it confirms that high IoU is not merely a result of experts reacting to prominent input features, but is rather a coordinated behavior orchestrated by the learned router. Experts no longer converge on which temporal regions matter because routing decisions become arbitrary. This is consistent with cross-layer routing agreement arising from learned routing behavior, not from the MoE architecture itself.

**Routing mechanism dictates stability profile.** Time-MoE exhibits a distinct Stability Gap: gentle perturbations rarely change routing ($<$1%), while aggressive perturbations succeed ($>$99%). In contrast, Moirai-MoE is sensitive, with routing changes occurring in 100% of inputs even under gentle noise. This divergence isolates the impact of the routing mechanism: Time-MoE's *learned gates* develop a decision margin that filters noise, whereas Moirai-MoE's *distance-based routing* (comparing queries to centroids) lacks this margin and responds to minute input variations. Note that while Moirai-MoE exhibits high *frequency* of change (100% of inputs), the *severity* is low: only $\sim$20% of individual expert selections change per input, compared to Time-MoE's near-zero change.

## 6 Discussion

Our experimental validation reveals properties of Time-MoE's routing architecture that have broader implications for understanding MoE models in time series forecasting.

**Ensemble Consensus vs. Expert Specialization.** A common assumption in MoE interpretability is that experts develop distinct specializations, such as one expert for trends and another for seasonality. Our saliency IoU analysis (mean 0.677) challenges this assumption: we observe that experts at different layers consistently identify the same temporal windows rather than partitioning the input space. This "verification committee" architecture may explain MoE robustness: multiple independent assessors attending to the same features provide a stable consensus even when individual assessments are noisy. This redundancy functions as a learned variance reduction mechanism; in noisy time series, multiple experts verifying the same temporal patterns allows the model to average out individual errors, producing more stable routing decisions.

**The Stability Gap.** The 300-fold difference between gentle (0.3%) and aggressive (99.7%) perturbation success rates reveals that Time-MoE's routing is robust. The router ignores superficial signal modifications and responds only when structural anchors, the patterns identified by saliency maps, are destroyed. This robustness is desirable for deployment but creates challenges for counterfactual-based explainability: meaningful "what-if" questions require substantial input modifications.

**Distributed Focus Scores.** The mean focus score of 0.563 indicates distributed attention across temporal windows. Rather than interpreting this as "low confidence," we understand it as reflecting the consensus architecture: when multiple experts attend to overlapping regions, the combined saliency is naturally distributed rather than concentrated on a single peak.

**Generalizability across architectures.** Our cross-architecture analysis reveals that Ensemble Consensus generalizes to other MoE models, though its strength varies with the routing mechanism. This finding suggests that consensus may be an emergent property of sparse expert selection under learned routing, rather than specific to Time-MoE's design. The absence of a Stability Gap in Moirai-MoE indicates that routing robustness is not inherent to MoE architectures but develops when gates learn to identify invariant signal features. These results have practical implications: practitioners seeking robust routing behavior should prefer learned gate mechanisms over distance-based approaches.

**Implications for MoE Interpretability.** Our findings suggest several reconsiderations for MoE explainability. First, interpretability frameworks should not assume expert specialization; our results indicate that MoE routing may rely on ensemble consensus rather than task partitioning. Second, for stable routing architectures, counterfactual methods face a tension: gentle perturbations that preserve signal semantics do not alter routing, while aggressive perturbations that change routing may destroy meaningful information. Third, rather than interpreting saliency maps as "importance scores," our framework positions them as identifying structural anchors, regions whose integrity is essential for routing stability.

**Residual stream and layer-specific effects.** The residual stream analysis (Section 5) reveals that saliency consensus is not fully explained by either the residual stream hypothesis or the pure learned-feature hypothesis. Perturbation effects propagate approximately uniformly through intermediate layers but the terminal layer (layer 11) shows distinctively heightened routing sensitivity. The non-monotonic IoU pattern across layer distances reflects which specific layer pairs are compared rather than a smooth function of distance. This nuanced picture, shared input propagation combined with layer-specific routing behavior, suggests that consensus arises from the interaction of architectural and learned components.

**Computational cost and optimization.** The per-explanation cost of 2–5 seconds for saliency and 3–8 seconds for counterfactual generation reflects the current unoptimized implementation. The dominant cost is sequential forward passes: multi-scale saliency at four window sizes requires approximately $4 \times \lceil T/s \rceil$ passes, and counterfactual generation evaluates up to 13 perturbation methods. Several concrete optimizations reduce this cost. First, perturbation trials within each saliency scale are independent, so batching multiple perturbed inputs into a single GPU forward pass reduces wall-clock time proportional to batch size. Second, baseline routing for a given input is computed once and reused across all expert analyses, saving $3\times$ redundant passes. Third, a reduced perturbation suite using only gentle and aggressive endpoints (2 methods instead of 13) provides coarse stability assessment in under 2 seconds per expert. Fourth, single-scale saliency at the medium window size ($w = 10$) gives a $4\times$ reduction in forward passes; the component ablation (Table 5) shows this configuration performs comparably to the full pipeline. For deployment, a tiered approach is practical: a fast mode (single-scale, 2 perturbation methods) completing in under 2 seconds per expert without GPU batching, and a full mode for offline diagnosis.

**Limitations.** Several limitations merit acknowledgment. Our cross-architecture comparison shows that Moirai-MoE differs from Time-MoE in multiple dimensions (routing mechanism, number of experts, layer count), making it difficult to isolate which factor causes the observed differences in stability profiles. Future work could examine models that vary only in routing mechanism while holding other factors constant. Additionally, the perturbations that successfully alter routing (zero-out, noise injection) may not correspond to semantically meaningful input modifications; future work could explore optimization-based counterfactuals that minimize semantic distance while achieving routing change. The component ablation study shows that individual components (multi-scale, adaptive windowing, multi-signal confidence) contribute incrementally rather than dramatically; the non-salient control experiment provides the strongest evidence for saliency map validity. The consensus–prediction correlation is dataset-dependent: a significant negative correlation appears for ETTh1, while other datasets show weaker effects, suggesting the relationship between routing consensus and prediction quality varies with data characteristics. Finally, our analysis operates at the input level; some routing decisions may be better explained by intermediate representations, and extending the framework to layer-wise attribution could address this limitation.

# 7 Conclusion and Future Work

We have presented RPATH, a post-hoc explainability framework for time series Mixture-of-Experts models that combines temporal saliency mapping, counterfactual generation, and uncertainty quantification. Our approach operates on frozen models without requiring gradient access or architectural modifications.

Experimental validation on Time-MoE-50M across 300 expert-sample pairs reveals two properties of the architecture. First, we observe *Ensemble Consensus*: experts at different layers consistently identify the same critical temporal windows (mean saliency IoU 0.677), challenging the assumption that MoE models achieve performance through distinct expert specializations. Second, we identify *Structural Robustness* through a Stability Gap, where gentle perturbations alter routing in only 0.3% of cases while aggressive perturbations succeed in 99.7%, indicating that routing decisions reflect structural anchors rather than superficial signal characteristics.

Together, these findings demonstrate that Time-MoE's performance stems from *Ensemble Redundancy*: multiple experts verify the same structural anchors, providing a robust consensus that is insensitive to noise but responsive to meaningful signal changes. Cross-architecture analysis on Moirai-MoE confirms that Ensemble Consensus is a general property of MoE models, though learned gate routing produces stronger consensus than distance-based approaches. The Stability Gap, however, appears specific to learned routing mechanisms. Our framework provides practitioners with tools to visualize expert attention, identify critical input regions, and quantify routing stability.

Several research directions emerge from this work. Understanding how ensemble consensus emerges during training could inform MoE architecture design and whether explicit consensus regularization improves robustness. If routing depends primarily on structural anchors, models might be compressed by reducing expert redundancy while preserving consensus coverage. Finally, the stability gap suggests a path toward certified routing robustness: inputs where gentle perturbations never alter routing may be flagged as high-confidence predictions. We provide a complete open-source implementation at `https://github.com/temex12/RPATH` to facilitate further research and practical applications in time series forecasting.

### Broader Impact Statement

RPATH is designed to support, not replace, human oversight of deployed time series Mixture-of-Experts models. By providing post-hoc explanations of routing decisions through temporal saliency and counterfactual analysis, the framework gives practitioners tools to inspect why specific experts are selected for a given input, information that is otherwise opaque in sparsely-routed architectures. This transparency matters most in domains where forecasting errors carry consequences, such as energy grid management, clinical decision support, and financial risk assessment, where stakeholders need to verify that model behavior aligns with domain knowledge before acting on predictions. We caution that explanations produced by RPATH characterize what input regions a frozen model relies upon, but do not certify that the underlying model is correct, fair, or appropriate for a given deployment context; explainability is a complement to, not a substitute for, validation against task-specific requirements and engagement with affected stakeholders. Practitioners should also be aware that the structural robustness we observe (the Stability Gap) means routing decisions resist superficial input modifications, which is desirable for noise tolerance but limits the use of gentle counterfactuals as a diagnostic tool, motivating the aggressive perturbation methods our framework includes.

### Author Contributions

T.M.A. conceived the RPATH framework, designed and implemented the methodology, conducted experiments, performed the analyses, and drafted the manuscript. Y.L. supervised the research, provided guidance on experimental design and analysis, and contributed to manuscript revisions. Both authors reviewed and approved the final version.

**Acknowledgments**

This work was supported by the Natural Sciences and Engineering Research Council of Canada (NSERC) [Alliance Grant #ALLRP 567562-2021 sponsored by British Columbia Energy Regulator (BC-ER)].

# A    Hyperparameter Sensitivity Analysis

We evaluate the robustness of RPATH's focus score to hyperparameter choices by varying (a) confidence signal weights $\alpha$ and (b) graduated validity breakpoints.

**Confidence signal weights.**    Table 13 reports focus score statistics and Spearman rank correlation $\rho$ with the default configuration across 8 weight vectors.

Table 13: Sensitivity of focus score ranking to confidence signal weights. All configurations preserve ranking ($\rho \geq 0.97$).

| Configuration | Mean Focus | Std | Spearman $\rho$ |
|---|---|---|---|
| default | 0.520 | 0.083 | 1.000 |
| equal | 0.506 | 0.083 | 1.000 |
| importance_dominated | 0.546 | 0.089 | 0.984 |
| sparsity_dominated | 0.496 | 0.079 | 0.983 |
| activation_dominated | 0.536 | 0.085 | 0.996 |
| consistency_dominated | 0.492 | 0.088 | 0.991 |
| no_consistency | 0.534 | 0.081 | 0.999 |
| no_sparsity | 0.550 | 0.093 | 0.970 |

**Held-out domain validation.**    To confirm that these weight choices generalize beyond the five datasets above, we repeat the analysis on 120 expert-sample pairs from two held-out domains: Electricity (energy consumption) and Traffic (road occupancy). Table 14 reports the results. All configurations again preserve ranking with $\rho \geq 0.98$ on the combined Electricity+Traffic set. Per-dataset, the minimum $\rho$ is 0.981 (Electricity) and 0.977 (Traffic), consistent with the ETT results ($\rho \geq 0.968$).

Table 14: Alpha sensitivity on held-out domains (120 pairs). Rankings remain stable ($\rho \geq 0.98$).

| Configuration | Mean Focus | Std | $\rho$ (E+T) | Min per-dataset $\rho$ |
|---|---|---|---|---|
| default | 0.599 | 0.180 | 1.000 | 1.000 |
| equal | 0.585 | 0.181 | 1.000 | 0.999 |
| importance_dominated | 0.611 | 0.177 | 0.995 | 0.993 |
| sparsity_dominated | 0.598 | 0.186 | 0.992 | 0.981 |
| activation_dominated | 0.611 | 0.178 | 0.999 | 0.998 |
| consistency_dominated | 0.558 | 0.180 | 0.996 | 0.994 |
| no_consistency | 0.619 | 0.180 | 0.999 | 0.999 |
| no_sparsity | 0.608 | 0.176 | 0.985 | 0.977 |

**Graduated validity breakpoints.**    Table 15 shows the effect of varying validity breakpoints. Absolute focus score levels change (as expected, since breakpoints directly scale the counterfactual validity component) but the core saliency rankings remain stable across configurations. The negative $\rho$ values reflect the focus score ranking (which includes the changed validity component), while the underlying saliency rankings are unchanged since saliency computation is independent of validity scoring.

Table 15: Sensitivity of focus score to validity breakpoint configurations.

| Configuration | Breakpoints | Mean Focus | Std | $\rho$ |
|---|---|---|---|---|
| default | 0.5/0.25/0.1 | 0.520 | 0.083 | 1.000 |
| strict | 0.6/0.4/0.2 | 0.639 | 0.033 | -0.121 |
| lenient | 0.4/0.15/0.05 | 0.739 | 0.033 | -0.121 |
| binary | 0.0/0.0/0.0 | 0.789 | 0.033 | -0.121 |

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
