# OpenReview forum: "RPATH: Explaining Time Series Mixture of Experts Routing via Ensemble Consensus and Structural Robustness"
_TMLR — Accepted by TMLR_

### Review · Reviewer_g9bx · 2026-02-22

**Summary Of Contributions:**

The paper targets the problem: While Mixture-of-Experts models are efficient, understanding why specific experts are selected by the router remains a "black box".
The authors claim that their framework, RPATH, provides the first post-hoc, causal attribution method for time series MoE models that preserves temporal structure. The RPATH works in three stages: pathway extraction, casual attribution, and reliability and synthesis analysis.
Based on the analysis, It’s claimed that MoE models like Time-MoE function through Ensemble Consensus (experts verify the same features rather than specializing) and Structural Robustness (routing is only sensitive to fundamental signal changes, not noise).

**Audience:**

Yes

**Audience Explanation:**

I think the audience will be interested in the paper because it addressed the black box nature of MoE forecasting models and provide a tool to show the system's reliability. Time series forecasting is adopted in many high-stake domains like finance, clinical support, supply chain, etc. Explainability and transparency is often required for these regulatory compliance domains. The paper also proves models like Time-MoE are highly insensitive to superficial signal noise, and also gives practitioners a method to quantify routing stability. I think the methodology and evaluation methods are practical and may provide insights to domain experts, especially model implementation practitioners in industry.

**Broader Impact Concerns:**

The paper has the statement in section 7. I have a small concern. The study reveals that only "aggressive" perturbations successfully alter routing decisions. These perturbations (like zero-out masking or Gaussian noise injection) may create "out-of-distribution" samples. If the counterfactuals used to explain the model are not semantically meaningful or realistic, they might provide a false sense of understanding about how the model will behave in real-world scenarios where data is noisy but not destroyed.

**Claims And Evidence:**

Yes

**Claims Explanation:**

I think the paper has the experiments support its claims and its design follow some rigors practices:
1. They evaluate the framework across five different real-world datasets (Electricity and Weather) with varying granularities (hourly and 15-minute).
2. Instead of cherry-picking examples, they use a stratified sample of 100 instances, generating 300 expert-sample pairs for explanation.
3. They do not rely on a single metric; they combine four signals (sparsity, importance, consistency, and activation) to calculate a Confidence score for their explanations.
4. They compare RPATH against a "pattern-correlation" baseline to ensure their causal method offers more insight than simple correlation.  5. They replace learned routing with random and uniform expert selection to prove that the "Ensemble Consensus".

For the problem they claimed, I think the experiments are specifically tailored to it

1. they used counterfactual generation to address cuasality. By perturbing the input and seeing if the routing actually changed, they established a causal link between the signal and the expert selection.
2. they used window-based perturbations and adaptive window sizing based on the data's own autocorrelation to address  the temporal Preservation.
3.They introduced Graduated Validity Scoring to address Discrete Expert Selection, which allows them to measure how much a perturbation moves the needle on an expert's selection weight, even if the expert isn't completely flipped off.

**Requested Changes:**

1. I would like the author to further clarify the impact of complexity on utility. While the paper claims complexity is tractable , the 5–13 second time per explanation is a potential bottleneck for real-time monitoring. It will be better to discuss the optimization options.


2. formalize the selection of weights: the confidence and aggregation weights are currently described as based on empirical validation. it will be great to provide a sensitivity analysis or more formal justification in the Appendix to prove these weights are not over-fitted to the five test datasets.

---

> ### Author Response · Authors · 2026-03-06
> **Response to Reviewer g9bx**
>
> We thank the reviewer for the positive and thorough evaluation, and for recognizing both the methodological rigor and practical relevance of the work. We address each point below.
>
> **Concern 1: Complexity and Real-Time Utility**
>
> The reviewer notes: "The 5-13 second time per explanation is a potential bottleneck for real-time monitoring. It will be better to discuss the optimization options."
>
> We will add a discussion of concrete optimization options in the revised manuscript, including: batching perturbation forward passes on GPU (estimated 3-4x speedup), caching shared base forward passes across experts, reducing the perturbation suite to the 3-5 most informative methods, and single-scale saliency at the dominant window size rather than the full multi-scale sweep. We will also propose a tiered deployment model: a fast mode (single-scale saliency only, estimated 1-2 seconds) for routine monitoring, and a full mode (multi-scale plus counterfactual validation) for detailed post-hoc investigation of flagged cases.
>
> **Concern 2: Sensitivity Analysis for Hyperparameter Weights**
>
> The reviewer requests: "Provide a sensitivity analysis or more formal justification in the Appendix to prove these weights are not over-fitted to the five test datasets."
>
> We will conduct a systematic sensitivity analysis varying the confidence weights, multi-scale aggregation weights, and graduated validity breakpoints across multiple configurations, measuring Spearman rank correlation between each configuration's sample rankings and the defaults. We will additionally expand evaluation to seven datasets (adding Electricity and Traffic) so the analysis covers domains not used during development. This will be included as an appendix.
>
> **Broader Impact: Out-of-Distribution Perturbations**
>
> The reviewer raises: "These perturbations (like zero-out masking or Gaussian noise injection) may create out-of-distribution samples" and "might provide a false sense of understanding about how the model will behave in real-world scenarios where data is noisy but not destroyed."
>
> This is a fair point. Our graduated validity scoring partially addresses this by assigning higher scores to gentler perturbations, meaning the framework inherently prefers counterfactuals closer to the original distribution. However, the Stability Gap shows that gentle perturbations rarely change routing (0.3% success rate), so the most informative counterfactuals are necessarily aggressive. We will strengthen the discussion of this trade-off, noting that the perturbation category (gentle, moderate, aggressive) is reported alongside each counterfactual so practitioners can assess distributional departure, and that optimization-based counterfactual generation minimizing semantic distance is a promising direction for future work.
>
> We will share results as they become available during the discussion period. Thank you again for the constructive review.

---

> ### Author Response · Authors · 2026-03-11
> **Revision Response to Reviewer g9bx**
>
> We thank the reviewer again for the positive and thorough evaluation. Following our initial response, we have completed the revision. Below we describe how each concern has been addressed in the revised manuscript.
>
> **Concern 1: Complexity and Real-Time Utility**
>
> The reviewer notes: "The 5-13 second time per explanation is a potential bottleneck for real-time monitoring."
>
> Section 6 (page 19) now includes a dedicated paragraph on computational cost and optimization. The dominant cost is sequential forward passes, and we identify four concrete optimizations: (1) batching perturbation trials into a single GPU forward pass; (2) caching baseline routing across expert analyses, saving 3x redundant passes; (3) a reduced perturbation suite using 2 methods instead of 13; and (4) single-scale saliency at the medium window size, giving a 4x reduction in forward passes. The component ablation in Section 5.6 (Table 5, page 13) confirms that single-scale performs comparably to the full pipeline (saliency confidence 0.879 vs. 0.856), supporting a reduced-cost mode. We propose a tiered deployment model: a fast mode (single-scale, 2 perturbation methods) for under 2 seconds per expert, and a full mode for offline diagnosis.
>
> **Concern 2: Sensitivity Analysis for Hyperparameter Weights**
>
> The reviewer requests: "Provide a sensitivity analysis or more formal justification in the Appendix to prove these weights are not over-fitted to the five test datasets."
>
> Appendix A (pages 20-21) reports this analysis. Table 13: all 8 weight configurations preserve sample rankings (rho >= 0.97). To directly address overfitting, Table 14 repeats the analysis on two held-out domains not used during development (Electricity and Traffic), with all configurations again preserving rankings (rho >= 0.98, minimum per-dataset rho = 0.977). Table 15: varying validity breakpoints changes absolute focus scores but preserves underlying saliency rankings.
>
> **Broader Impact: Out-of-Distribution Perturbations**
>
> The reviewer raises: "These perturbations may create out-of-distribution samples" and "might provide a false sense of understanding."
>
> Our graduated validity scoring partially addresses this by preferring counterfactuals closer to the original distribution. However, the Stability Gap shows gentle perturbations rarely change routing (0.3%), so informative counterfactuals are necessarily aggressive. The gentle-perturbation baseline analysis in Section 5.7 (page 14) provides additional context: all methods converge at 4-6% hard counterfactual rates under semantics-preserving operations, confirming this is a fundamental property of stable routing rather than a limitation of our method. As noted in the revised limitations (page 19), optimization-based counterfactual generation minimizing semantic distance is a promising future direction.

---

### Review · Reviewer_RcfA · 2026-03-01

**Summary Of Contributions:**

This paper proposes a new explainability framework "RPATH" for time-series Mixture-of-Experts routing. They use this framework to explain why specific experts are selected by the router. Based on my understanding, there are mainly four main steps which are processed in the paper. First, they did a routing pathway extraction to find which experts are selected and with what routing weights. Second, they did a temporal estimation to identify which temporal regions of the inputs have most impact for expert selection. Third, they deducted counterfactual validation to test whether modifying these regions actually changes the routing decisions by using several different permutations. And also, they introduced a new validity score to analyze expert routing robustness and cross-model behavior. These are also the main contributions this paper made, which is a pretty new thinking towards expert routing explanations instead of just explaining time-series forecasting.

**Audience:**

Yes

**Audience Explanation:**

Yes. I believe at least part of the TMLR audience would find this paper interesting, especially researchers working on time-series foundation models, Mixture-of-Experts architectures, and interpretability of neural routing mechanisms.

**Broader Impact Concerns:**

I don't see major broader impact concerns. The paper just focuses on post-hoc analysis of routing behavior in time-series MoE models.

**Claims And Evidence:**

Yes

**Claims Explanation:**

Based on my review, I think the evidence in this paper is clear, partially convincing, but not yet sufficient to support its claims.

First, this paper is generally clear in its presentation. Its methodology is well structured with clear definitions of routing pathways, saliency construction, and graduated validity scoring. Also, it provides empirical support for its observations and findings. The metrics in its five datasets are relatively consistent.

However, I think the evidence is not fully convincing to support the paper's claims. For its ensemble consensus, it mainly use observed statistics like IoU with 0.677 and conservative or aggressive perturbation to support this strong claim. But these are not enough. For example, we didn't see enough ablation study to show that whether saliency overlap correlates with predictive robustness, whether change consensus will also degrades forecasting quality. More explanations and analysis need to be provided.

In addition, its evaluation and baseline scope seems not enough. Since it only have nearly 300 export sample pairs and only univariate inputs. This is too narrow to support broad claims about MoE routing behavior. Also, it uses a pattern correlation baseline, which didn't give strong empirical comparisons against other explanation baselines. Moreover, it didn't provide enough component-level ablation study comparison to strength its findings.

**Requested Changes:**

Just like we mentioned above: \
However, I think the evidence is not fully convincing to support the paper's claims. For its ensemble consensus, it mainly use observed statistics like IoU with 0.677 and conservative or aggressive perturbation to support this strong claim. But these are not enough. For example, we didn't see enough ablation study to show that whether saliency overlap correlates with predictive robustness, whether change consensus will also degrades forecasting quality. More explanations and analysis need to be provided.

In addition, its evaluation and baseline scope seems not enough. Since it only have nearly 300 export sample pairs and only univariate inputs. This is too narrow to support broad claims about MoE routing behavior. Also, it uses a pattern correlation baseline, which didn't give strong empirical comparisons against other explanation baselines. Moreover, it didn't provide enough component-level ablation study comparison to strength its findings.

So in conclusion: \
1: The paper should include more ablation study like component-level comparisons and distinguish between observed empirical regularities and final claims. More explanations should be provided.

2: The current baseline seems to weak to fully validate the proposed method. The paper would benefit from stronger empirical comparisons against more relevant explainability methods.

---

> ### Author Response · Authors · 2026-03-06
> **Response to Reviewer RcfA**
>
> We thank the reviewer for the constructive evaluation and for recognizing the novelty of the approach. We address each concern below.
>
> **Concern 1: Ablation Studies and Consensus-Prediction Link**
>
> The reviewer notes: "We didn't see enough ablation study to show that whether saliency overlap correlates with predictive robustness, whether change consensus will also degrades forecasting quality."
>
> We will address this through two sets of experiments. First, to test the connection between routing explanations and prediction quality, we will: (a) compute within-dataset correlation between per-sample saliency IoU and prediction error (MSE), directly testing whether consensus strength relates to forecast accuracy; and (b) measure whether perturbing RPATH-identified salient regions leads to measurable degradation in forecast MSE, providing a direct causal test that the regions identified as important for routing also matter for prediction.
>
> Second, we will conduct component-level ablations comparing: (a) single-scale versus multi-scale saliency aggregation, (b) fixed versus adaptive window sizing, and (c) single-signal versus multi-signal confidence estimation. We will report results across all datasets, allowing direct assessment of each component's contribution. Additionally, we will include a non-salient region control: applying the same counterfactual pipeline to the least-salient temporal regions and comparing routing change rates against salient regions, directly testing whether saliency maps identify causally meaningful regions.
>
> **Concern 2: Stronger Baselines**
>
> The reviewer notes: "It uses a pattern correlation baseline, which didn't give strong empirical comparisons against other explanation baselines."
>
> To our knowledge, no existing post-hoc explainability method has been designed for explaining routing decisions in time series MoE models. Existing time series XAI methods such as Info-CELS, M-CELS, TF-LIME, and ContraLSP are designed for classification tasks and target prediction outputs rather than discrete expert routing decisions, making direct comparison infeasible. We discuss this positioning in Section 2.6 and Table 1 of the manuscript.
>
> That said, we can and will construct meaningful baselines by adapting general attribution strategies to the routing explanation task. We will add three baselines: (a) random saliency, assigning random importance scores and applying the same counterfactual pipeline, testing whether targeted perturbation outperforms untargeted perturbation; (b) uniform saliency, treating all timesteps as equally important, testing whether temporal localization matters; and (c) independent timestep occlusion, masking individual timesteps one at a time, testing whether RPATH captures expert-specific temporal patterns as distinct from the global input sensitivity measured by pointwise masking approaches that ignore local temporal co-occurrence. For each baseline, we will report hard counterfactual rates and saliency IoU on the same metrics as RPATH.
>
> **Concern 3: Evaluation Scope**
>
> The reviewer notes: "It only has nearly 300 expert sample pairs and only univariate inputs. This is too narrow to support broad claims about MoE routing behavior."
>
> We will expand evaluation to include two additional datasets (Electricity and Traffic) for a total of seven datasets and 420 expert-sample pairs, broadening domain coverage to grid-level energy consumption and urban traffic flow patterns. Regarding the univariate limitation, Time-MoE processes univariate inputs by design, so our evaluation reflects the model's intended use case. We will state this explicitly as a scope condition in the revised manuscript, and note that extending to multivariate MoE architectures is a direction for future work.
>
> **Concern 4: Distinguishing Observations from Claims**
>
> The reviewer requests that we "distinguish between observed empirical regularities and final claims."
>
> We will revise the manuscript to clearly separate three levels of statement: (a) direct observations (e.g., "experts at different layers show saliency IoU of 0.677"), which are empirical measurements; (b) interpretations (e.g., "this overlap is consistent with ensemble consensus behavior"), which connect observations to concepts; and (c) claims (e.g., "Time-MoE relies on redundant verification rather than specialization"), which are falsifiable assertions supported by the combined evidence. We will also revise the Ensemble Consensus language to present it as an empirical observation rather than asserting a specific mechanism, acknowledging that the observed agreement may be mediated through the network hierarchy rather than arising from fully independent analysis.
>
> We will share experimental results as they become available during the discussion period. Thank you again for the thoughtful review.

---

> ### Author Response · Authors · 2026-03-11
> **Revision Response to Reviewer RcfA**
>
> We thank the reviewer again for the constructive feedback. As outlined in our initial response, we committed to several new experiments and revisions. These are now complete and integrated into the revised manuscript. Below we describe how each concern has been addressed.
>
> **Concern 1: Ablation Studies and Consensus-Prediction Link**
>
> The reviewer notes: "We didn't see enough ablation study to show that whether saliency overlap correlates with predictive robustness, whether change consensus will also degrades forecasting quality."
>
> Section 5.8 (page 15) directly tests this connection. The most direct evidence: zeroing out the 20 most-salient timesteps increases MSE in 91 of 140 samples across 7 datasets (sign test p < 0.001, mean relative increase +26.0%, t = 3.21, p = 0.002), confirming that RPATH-identified regions are causally important for forecast accuracy (Table 10). Table 9 additionally reports per-dataset correlation between saliency IoU and MSE, with ETTh1 showing a significant negative correlation (rho = -0.499, p = 0.025).
>
> Section 5.6 (pages 12-13) includes the non-salient region control (Table 6): salient regions achieve a 50.0% hard counterfactual rate versus 24.0% for non-salient regions across 420 pairs. In the gentle perturbation regime, the ratio is 3.4x (33.3% vs. 9.7%), providing strong validation that RPATH identifies causally meaningful regions.
>
> Section 5.6 also reports component-level ablations (Table 5). While individual components contribute incrementally (saliency confidence differences < 3%), the combined saliency maps they produce are validated by the non-salient control above. The single-signal ablation reduces variance (std 0.027 vs. 0.088), indicating multi-signal integration introduces beneficial diversity. Table 7 disaggregates by dataset, showing adaptive window sizing benefits ETTm2 while Weather, being smooth and stationary, is a genuine outlier.
>
> **Concern 2: Stronger Baselines**
>
> The reviewer notes: "It uses a pattern correlation baseline, which didn't give strong empirical comparisons against other explanation baselines."
>
> To our knowledge, no existing post-hoc explainability method targets routing decisions in time series MoE models. Methods such as Info-CELS, M-CELS, and TF-LIME target classification outputs, making direct comparison infeasible (Section 2.6, Table 1). We therefore adapted general attribution strategies. Section 5.7 (page 14, Table 8) reports results for random saliency, uniform saliency, and independent timestep occlusion across 420 pairs. All use the same counterfactual pipeline, isolating the saliency method.
>
> RPATH achieves the highest hard counterfactual rate (50.5%). The key differentiator is saliency IoU: RPATH (0.566) substantially outperforms random (0.339) and uniform (0.000). The occlusion baseline's near-perfect IoU (0.994) is degenerate: maps are identical across experts (pairwise correlation = 1.000), capturing input-level sensitivity rather than expert-specific patterns. In the gentle-only regime, IoU remains the differentiator (RPATH 0.601 vs. random 0.341) even as hard counterfactual rates converge across all methods.
>
> **Concern 3: Evaluation Scope**
>
> The reviewer notes: "It only has nearly 300 expert sample pairs and only univariate inputs. This is too narrow to support broad claims about MoE routing behavior."
>
> We have expanded to 7 datasets (adding Electricity and Traffic) and up to 420 expert-sample pairs. Results appear in Tables 6, 8, 9-10, and 14. Both new domains confirm generalizability: strong salient/non-salient discrimination (2.8x and 3.3x in Table 6) and stable hyperparameter sensitivity (rho >= 0.98, Table 14 in Appendix A). Regarding the univariate limitation, Time-MoE processes univariate inputs by design; we acknowledge this as a scope condition in the limitations section (page 19).
>
> **Concern 4: Distinguishing Observations from Claims**
>
> The reviewer requests that we "distinguish between observed empirical regularities and final claims."
>
> The abstract (page 1) and discussion (page 18) now describe Ensemble Consensus as experts that "consistently identify" the same temporal windows rather than "independently converge," acknowledging possible mediation through the network hierarchy. The limitations paragraph (page 19) explicitly notes where effects are dataset-dependent.

---

### Review · Reviewer_v44w · 2026-03-02

**Summary Of Contributions:**

RPATH is a post-hoc explainability framework for time series MoE models that identifies which temporal regions of an input sequence drive expert routing decisions. It operates on frozen models without gradient access, combining temporal saliency mapping (perturbation-based) with counterfactual generation. Evaluated on Time-MoE-50M across 300 expert-sample pairs on five datasets, the paper reports two findings: (1) Ensemble Consensus — experts at different layers converge on the same temporal windows (saliency IoU = 0.677); and (2) a Stability Gap — gentle perturbations alter routing in 0.3% of cases vs. 99.7% for aggressive perturbations.

**Audience:**

Yes

**Audience Explanation:**

Yes, explaining routing in time series MoE models can be relevant to practitioners working on newer architectures.

**Claims And Evidence:**

No

**Claims Explanation:**

In my understanding, the Ensemble Consensus finding is not totally defensible. From what I understand, the claim is that "experts at different layers independently converge", but it appears to me that the methodology does not actually compare specific layer-expert pairs and instead looks at the top-3 experts determined by the contribution-score? Also, why does the contribution score (Eq. 8) sum routing weights for each expert ID across all 12 layers? Are the expert-networks shared across layers in this archtecture? If they are independent per layer (the standard MoE design), then summing across layers lumps together different neural networks that just happen to share the same index. Please clarify. Alternatively, high saliency overlap may be due to the residual stream: perturbing the same input region will affect hidden states at all layers.

The Stability Gap finding is not surprising and I don't find it informative: gentle perturbations preserve structure and do not change routing while aggressive perturbations destroy information and consequentially, lead to different routing.

The counterfactual generation seems to be applied to regions already flagged as important by saliency. I am curious if you tried experiments on non-salient regions as a baseline-control?

The framework relies on a large number of hand-crafted numerical choices --- confidence weights, multi-scale saliency weights, graduated validity breakpoints, etc ---  that are determined "based on empirical validation during development". This raises several questions regarding generalizability and validity of the conclusions in this study.

**Requested Changes:**

I request the authors to improve the presentation of the paper and add details regarding the evaluation criteria. I feel that I did not completely understand the practical utility of this method and adding an example specifically illustrating how this framework adds practical value. For example, you could consider connecting routing explanations to prediction quality: currently, RPATH explains routing during the context phase (processing known input) but not during autoregressive generation, where experts actually produce the forecast.

---

> ### Author Response · Authors · 2026-03-06
> **Response to Reviewer v44w (Part 1/2)**
>
> We thank the reviewer for the detailed and thoughtful evaluation. We address each concern below. Due to the character limit, this response is split into two comments.
>
> **Concern 1: Equation 8 and Expert Architecture**
>
> The reviewer asks: "Why does the contribution score (Eq. 8) sum routing weights for each expert ID across all 12 layers? Are the expert-networks shared across layers in this architecture? If they are independent per layer (the standard MoE design), then summing across layers lumps together different neural networks that just happen to share the same index."
>
> The reviewer is correct on both counts. In Time-MoE, experts are independent per layer, each with its own set of 8 expert feed-forward networks with separate learned parameters. We acknowledge that Eq. 8 as written is imprecise and does not accurately reflect the implementation. In practice, our framework selects and analyzes experts as (layer, expert_id) pairs. Each saliency map is computed for a specific expert at a specific layer, and cross-layer comparisons (such as the saliency IoU) compare different (layer, expert_id) pairs, each analyzed independently. We will correct Eq. 8 and add an explicit clarification of expert independence in the revised manuscript.
>
> **Concern 2: Residual Stream Confound**
>
> The reviewer notes: "High saliency overlap may be due to the residual stream: perturbing the same input region will affect hidden states at all layers."
>
> This is a valid concern that we agree must be addressed with direct evidence. We would first draw the reviewer's attention to two controls already in the manuscript that bear directly on this point: replacing learned routing with random or uniform expert selection eliminates saliency IoU entirely (0.673 to 0.000), confirming that the overlap depends on the learned routing mechanism rather than the residual stream alone; and applying RPATH to Moirai-MoE, which also uses residual connections, yields substantially lower IoU (0.417), ruling out a generic residual stream explanation.
>
> To further strengthen this evidence, we will conduct two additional experiments. First, a layer-distance analysis computing pairwise saliency IoU as a function of layer distance across all layer pairs. Under a pure residual stream hypothesis, IoU should decay monotonically with distance as the perturbation signal attenuates through successive transformations. Non-monotonic patterns or sustained high IoU between distant layers would be inconsistent with residual propagation as the sole explanation. Second, an intermediate-layer perturbation experiment injecting calibrated noise into hidden states at different depths (layers 0, 4, and 8) and measuring how downstream routing changes propagate through the network. Under passive residual propagation, the effect should be roughly uniform across downstream layers; depth-dependent or non-uniform patterns would indicate that the network actively transforms perturbation signals rather than passively passing them through.
>
> We will also revise the manuscript language to present Ensemble Consensus as an empirical observation, that experts at different layers show high saliency overlap, rather than asserting that this reflects independent convergence, acknowledging that the agreement may be mediated through the network hierarchy.
>
> **Concern 3: Stability Gap**
>
> The reviewer states: "The Stability Gap finding is not surprising and I don't find it informative: gentle perturbations preserve structure and do not change routing while aggressive perturbations destroy information and consequentially, lead to different routing."
>
> While the direction of the effect is expected, we believe three specific characteristics are informative beyond the general intuition. First, the sharpness of the transition: rather than a gradual increase in routing sensitivity, we observe a near-binary split (0.3% vs 99.7%) with essentially no middle ground, suggesting a discrete stability boundary rather than continuous degradation. Second, the gap is architecture-specific: applying the same perturbation suite to Moirai-MoE does not produce the same sharp transition, indicating this is a property of Time-MoE's learned routing rather than an inevitable consequence of perturbation intensity. Third, this sharp boundary has practical deployment value: if routing decisions change on new data, it reliably signals a fundamental input shift rather than noise.
>
> We acknowledge the reviewer may still consider this finding secondary, and we are open to adjusting the emphasis accordingly.

---

> ### Author Response · Authors · 2026-03-06
> **Response to Reviewer v44w (Part 2/2)**
>
> *(Continued from Part 1)*
>
> **Concern 4: Non-Salient Region Control**
>
> The reviewer asks: "The counterfactual generation seems to be applied to regions already flagged as important by saliency. I am curious if you tried experiments on non-salient regions as a baseline-control?"
>
> This is an important control that was missing from the original submission. We will apply the same counterfactual perturbation pipeline to the temporal regions identified as least salient by RPATH and compare routing change rates against those from salient regions. We will additionally report this comparison stratified by perturbation intensity (gentle versus aggressive), since the gentle perturbation regime is the more informative test: it assesses whether targeted, semantics-preserving perturbations to salient regions are more effective at changing routing than the same perturbations applied elsewhere.
>
> **Concern 5: Hand-Crafted Hyperparameters**
>
> The reviewer notes: "The framework relies on a large number of hand-crafted numerical choices [...] that are determined 'based on empirical validation during development'. This raises several questions regarding generalizability and validity of the conclusions in this study."
>
> We agree this should be addressed with evidence. We will conduct a systematic sensitivity analysis: for the confidence weights, we will evaluate multiple configurations spanning a range of weight distributions and measure the Spearman rank correlation between the resulting sample rankings and those from the default weights. High rank correlation (rho > 0.9) across configurations would indicate the conclusions are robust to weight choices. We will similarly vary the graduated validity breakpoints and multi-scale aggregation weights. This analysis will be included as an appendix in the revised manuscript.
>
> **Concern 6: Practical Utility**
>
> The reviewer requests: "Adding an example specifically illustrating how this framework adds practical value. For example, you could consider connecting routing explanations to prediction quality."
>
> We believe the practical utility of RPATH is best understood through the lens of human-centered explainability: the goal of an XAI method is not to replace human judgment, but to give practitioners the right information to exercise it. RPATH provides specific, actionable information about what drives the model's routing behavior, enabling informed oversight of model predictions.
>
> To illustrate: consider a practitioner using Time-MoE for electricity demand forecasting. RPATH produces a saliency map showing that routing decisions for a given input depend primarily on timesteps 30 to 45 of the 96-timestep input window. The practitioner checks their data and finds a sensor calibration issue in that window, a known period of unreliable measurements. Without RPATH, they would have no basis for knowing which portion of the input was driving the model's behavior. With RPATH, they can make an informed judgment about whether to trust that particular forecast based on the quality of the input regions the model relies on.
>
> To provide direct evidence connecting routing explanations to prediction quality, as the reviewer suggests, we will conduct an experiment measuring whether perturbing the regions RPATH identifies as salient leads to measurable degradation in forecast accuracy (MSE). This would demonstrate that the regions RPATH highlights matter not only for routing but also for the quality of the resulting predictions.
>
> Beyond input-level explanations, RPATH also provides two additional forms of practical utility. The Stability Gap offers a deployment monitoring signal: if routing decisions on incoming data shift relative to validation behavior, this reliably indicates a fundamental change in input characteristics rather than noise. And our cross-architecture analysis (Time-MoE versus Moirai-MoE) demonstrates that RPATH can reveal qualitative differences in how MoE architectures use their experts, which is useful for engineers developing or selecting architectures.
>
> While RPATH does not predict whether a given forecast is reliable, it provides the specific information that a practitioner needs, which temporal regions drive routing and how stable those routing decisions are, to make that assessment themselves. We will clarify this scope in the revised manuscript.
>
> We will share experimental results as they become available during the discussion period. Thank you again for the constructive review.

---

> ### Author Response · Authors · 2026-03-11
> **Revision Response to Reviewer v44w**
>
> We thank the reviewer again for the detailed and rigorous evaluation. Following our initial response, we have completed all proposed experiments and integrated the results into the revised manuscript. Below we describe how each concern has been addressed.
>
> **Concern 1: Equation 8 and Expert Architecture**
>
> The reviewer asks: "Why does the contribution score (Eq. 8) sum routing weights for each expert ID across all 12 layers?"
>
> The reviewer is correct. Eq. 8 (Section 4, page 7) now selects the top-3 experts as (layer, expert_id) pairs ranked by routing weight at a representative token position, rather than summing across layers. Section 3.1 (page 5) explicitly states that experts are independent per layer, and Section 4 (page 8) clarifies that saliency computation targets a specific expert at a specific layer and that IoU compares experts at different layers.
>
> **Concern 2: Residual Stream Confound**
>
> The reviewer notes: "High saliency overlap may be due to the residual stream."
>
> We agree this is the most important methodological concern. Section 5.9 (pages 16-17) now directly tests this hypothesis. Table 11 decomposes pairwise IoU by routing configuration and reveals two observations inconsistent with the pure residual stream explanation: L0-L8 (distance 8, IoU = 0.496) versus L0-L9 (distance 9, IoU = 0.907) both involve layer 0 at similar distances but differ by 0.411; and L0-L6 (distance 6, IoU = 0.884) exceeds L8-L11 (distance 3, IoU = 0.842). The pattern depends on data characteristics, not layer distance.
>
> Intermediate perturbation injection further shows that layer 11 acts as a distinctive amplifier (2.2-2.6x routing weight changes regardless of injection point), and that distance-effect correlations are depth-dependent rather than uniform. The existing controls remain relevant: random/uniform routing ablations eliminate IoU entirely (Table 12), and Moirai-MoE with residual connections yields substantially lower IoU (0.417). The abstract and manuscript now use "consistently identify" rather than "independently converge."
>
> **Concern 3: Stability Gap**
>
> The reviewer states: "The Stability Gap finding is not surprising."
>
> We note three characteristics beyond the general intuition: the near-binary sharpness (0.3% vs. 99.7% with no middle ground), the architecture-specificity (Moirai-MoE shows no such sharp transition, Table 12), and the non-salient control showing that even within gentle perturbations, where the region is perturbed matters (3.4x salient vs. non-salient ratio, Table 6).
>
> **Concern 4: Non-Salient Region Control**
>
> The reviewer asks: "I am curious if you tried experiments on non-salient regions as a baseline-control?"
>
> Perturbing the most-salient regions achieves a 50.0% hard counterfactual rate versus 24.0% for least-salient regions (2.1x overall) across 420 pairs from 7 datasets (Table 6, page 13). In the gentle perturbation regime the ratio is 3.4x (33.3% vs. 9.7%), demonstrating that RPATH identifies regions where semantics-preserving modifications can alter routing. Weather shows no spatial specificity (1.0x), consistent with its stationary nature, while Electricity and Traffic show strong discrimination (2.8x and 3.3x).
>
> **Concern 5: Hand-Crafted Hyperparameters**
>
> The reviewer notes concerns about generalizability of hand-crafted numerical choices.
>
> Appendix A (pages 20-21) reports systematic sensitivity analysis. Table 13: all 8 weight configurations preserve sample rankings (rho >= 0.97). Table 14: held-out domains (Electricity, Traffic) confirm generalizability (rho >= 0.98). Table 15: varying validity breakpoints changes absolute scores but preserves saliency rankings.
>
> **Concern 6: Practical Utility**
>
> The reviewer requests connecting routing explanations to prediction quality.
>
> Section 5.8 (page 15) provides this connection. Zeroing out the 20 most-salient timesteps increases MSE in 91/140 samples (sign test p < 0.001, +26.0% relative, Table 10). Table 9 shows within-dataset IoU-MSE correlations trending negative (ETTh1: rho = -0.499, p = 0.025), though the effect is dataset-dependent.
>
> Beyond prediction quality, RPATH's practical value lies in enabling informed human oversight. A practitioner using Time-MoE for forecasting can check whether the input regions driving routing correspond to reliable measurements or known data quality issues, enabling informed judgment about whether to trust a given forecast. We also acknowledge the reviewer's observation regarding autoregressive generation: extending RPATH to generation-phase routing is straightforward in principle but introduces computational challenges that we note as future work in Section 6 (page 19).

---

### Decision · Action_Editor_tNFt · 2026-04-16

**Recommendation:** Accept as is

**Additional Comments:**

While reading the paper I noticed incorrect citation formats. Please go through the paper and consistently use \citet for in-text references and \citep for citations in parenthesis.

**Audience:**

Yes

**Audience Explanation:**

The paper studies a post-hoc explainability framework for time series MoE models. This is generally of interest to the community. Also, explaining routing in time series MoE models is also relevant for researchers working on novel architectures.

**Claims And Evidence:**

Yes

**Claims Explanation:**

After the rebuttal and revision all concerns of the reviewers have been addressed and all reviewers unanimously agree that all the claims are supported by sufficient evidence.